# Enhancing Graph Representations with Neighborhood-Contextualized Message-Passing

**Brian Godwin Lim**                                    *brian.lim@naist.ac.jp*
*Nara Institute of Science and Technology, Kyoto University, and Ateneo de Manila University*

**Galvin Brice Lim**                                    *galvinlim@pic.net.ph*
*UNI-President Information Philippines Corporation*

**Renzo Roel Tan**                                      *rr.tan@is.naist.jp*
*Nara Institute of Science and Technology and Ateneo de Manila University*

**Irwin King**                                          *king@cse.cuhk.edu.hk*
*The Chinese University of Hong Kong*

**Kazushi Ikeda**                                       *kazushi@is.naist.jp*
*Nara Institute of Science and Technology*

**Reviewed on OpenReview:** *https://openreview.net/forum?id=6nxdSt5pSn*

## Abstract

Graph neural networks (GNNs) have become an indispensable tool for analyzing relational data. Classical GNNs are broadly classified into three variants: convolutional, attentional, and message-passing. While the standard message-passing variant is expressive, its typical *pair-wise* messages only consider the features of the center node and each neighboring node *individually*. This design fails to incorporate contextual information contained within the *broader* local neighborhood, potentially hindering its ability to learn meaningful relationships within the *entire* set of neighboring nodes—a critical limitation for complex domains like financial network anomaly detection and molecular property prediction. To address this, the paper first refines the concept of neighborhood-contextualization within GNNs, leveraging ideas from set-based aggregation methods and a key property of the attentional variant. This then serves as the basis for generalizing the message-passing variant to the proposed neighborhood-contextualized message-passing (NCMP) framework. To demonstrate its utility, a simple, mathematically grounded method to parametrize and operationalize NCMP is presented, leading to the development of the proposed Soft-Isomorphic Neighborhood-Contextualized Graph Convolution Network (SINC-GCN). Across a diverse set of synthetic and benchmark datasets, SINC-GCN strikes a highly favorable balance between expressivity and efficiency. Notably, while more complex models incur significant computational overhead, SINC-GCN delivers substantial performance gains with considerable effect sizes over baseline GNN models while maintaining a highly efficient asymptotic runtime complexity, further underscoring the distinctive utility of neighborhood-contextualization. Overall, by integrating multiset neighborhood context, the proposed NCMP framework serves as a practical and scalable path toward enhancing the graph representational power of classical GNNs.

# 1 Introduction

In the modern age of big data, graphs have become an indispensable tool for modeling complex relationships. Many real-world systems may be naturally represented as graphs, where nodes represent entities and edges represent interactions. For instance, financial systems may be viewed as graphs of users connected via transactions, in which the aggregate local interaction is crucial for detecting anomalous behaviors. Moreover, social networking sites may correspond to graphs of people connected through friendships, where the immediate network of connections provides critical information on user characteristics. Similarly, molecules may be represented as graphs of atoms connected by chemical bonds, with the local atomic composition playing a vital role in the overall chemical structure. Furthermore, centuries of research in the field of graph theory have provided a rich set of mathematical tools to study and analyze these structures.

With the growing interest in the field of machine learning from both academia and industry, graph neural networks (GNNs) have emerged as a special subclass of deep learning architectures specifically designed to process graph-structured data. In contrast to traditional architectures, GNNs consider both the graph structure via edge connections and the information contained within the nodes, making them well-suited for various graph tasks. For example, they may be used for node property prediction (*e.g.*, identifying suspicious users in financial systems based on anomalies in transaction patterns), edge prediction (*e.g.*, suggesting friends in social networking sites based on similarities in characteristics), and graph property prediction (*e.g.*, predicting chemical properties of molecules based on atomic composition).

In the literature, one-hop localized GNN architectures, which are the primary focus of this paper, may be broadly classified into three variants or *flavors*: convolutional, attentional, and message-passing. Foundational works in the field, rooted in spectral graph theory, mainly fall under the convolutional variant, whereby each node aggregates information or messages from its neighboring nodes by simply considering each neighborhood feature individually. With the introduction of the Transformer, various works have adapted the attention mechanism into GNNs, whereby each node aggregates messages from its neighboring nodes, similarly considering each neighborhood feature individually, with a dynamic weighting scheme based on their relative importance. More recently, with the development of hardware, many works have studied message-passing variants to push the limits of GNNs, whereby each node aggregates messages from its neighboring nodes by considering both its own features and the features of each neighbor. Within this paradigm, researchers agree that the attentional variant is more expressive than the convolutional variant in terms of graph representational power, as the latter may be expressed as a particular instance of the former. Moreover, the message-passing variant is largely agreed to be the most expressive GNN variant, as it can be thought of as a generalization of the other two variants.

Despite the success and wide adoption of the classic message-passing variant, it has a key architectural limitation: the *pair-wise* messages are traditionally calculated using only the features of the center node and each *individual* neighboring node. Crucially, this design overlooks the rich contextual information embedded in the *broader* context of the local neighborhood, specifically with the relationships among the *entire* set of neighboring nodes—a critical limitation for complex GNN applications such as those described previously. In line with this key insight, this work:

1. Refines the concept of **neighborhood-contextualization** within GNNs, leveraging established set-based aggregation literature (Zaheer et al., 2017; Qi et al., 2017) and an implicit yet crucial property of the attentional variant;

2. Proposes the **neighborhood-contextualized message-passing (NCMP) framework** as a practical generalization of the message-passing variant, featuring both contextualized messages, as defined in Lim et al. (2025a), and neighborhood-contextualization; and

3. Presents a theoretical discussion on one simple, mathematically grounded method for its parametrization and operationalization, leading to the development of the **Soft-Isomorphic Neighborhood-Contextualized Graph Convolution Network (SINC-GCN)**.

Through extensive evaluation in both synthetic and benchmark datasets across node and graph property prediction tasks, SINC-GCN strikes a balance between performance and efficiency, achieving consistent and substantial gains against baseline GNN models. Overall, the NCMP framework and its SINC-GCN instantiation offer a theoretically-grounded and practical path toward enhancing the representational capability of classical GNNs by integrating local structural semantics in a computationally efficient manner.

The paper is organized as follows. Section 2 first presents an overview of the classical GNNs. Section 3 subsequently motivates the proposed NCMP framework and presents a theoretical discussion for developing the simple SINC-GCN instance. Section 4 then highlights their practical utility and expressivity with experiments on synthetic and benchmark datasets. Section 5 finally concludes with a summary of the contributions and recommendations for future work.

## 2 Graph Neural Networks

Let $\mathcal{G} = (\mathcal{V}, \mathcal{E})$ be a graph, with $\mathcal{N}(u) \subseteq \mathcal{V}$ denoting the set of nodes adjacent to node $u \in \mathcal{V}$ and $\boldsymbol{h_u}$ denoting the features of node $u$. In the literature, the development of classical one-hop localized GNNs generally follows the chronology of convolutional, attentional, and message-passing variants. For brevity, only the core message-passing operations defining the GNN architectures are presented in the subsequent discussion.

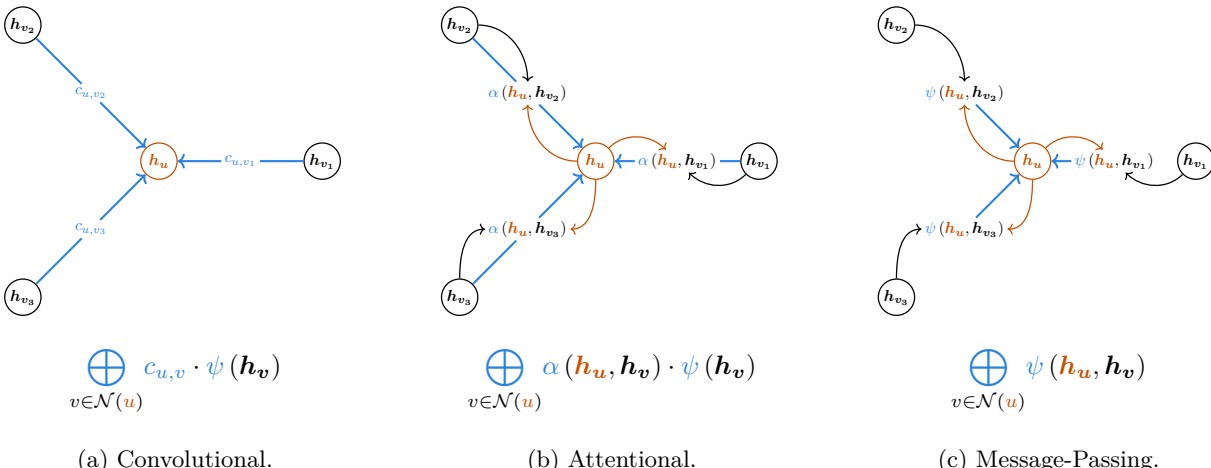

(a) Convolutional.  (b) Attentional.  (c) Message-Passing.

Figure 1: Graph Neural Network Architecture Variants.

### 2.1 Convolutional Variant

Early works in graph machine learning attempted to define the convolution operation on graphs by building upon spectral graph theory, often using a graph Fourier transform on the graph Laplacian. However, the computational complexity of calculating the full spectrum led to the development of more efficient polynomial approximations. Among these works was the Graph Convolution Network (GCN), introduced as a learnable, first-order approximation of the graph convolution localized to the one-hop neighborhood, defined as

$$\boldsymbol{h_u^*} = \sum_{v \in \mathcal{N}(u)} \frac{1}{\sqrt{|\mathcal{N}(u)|}\sqrt{|\mathcal{N}(v)|}} \, \boldsymbol{W h_v}, \tag{1}$$

where $\boldsymbol{W}$ is a learnable linear transformation and $\boldsymbol{h_u^*}$ is the updated features for node $u$ after the convolution operation (Kipf & Welling, 2017). GCN was shown to outperform existing methods in transductive semi-supervised tasks. Contemporaneously, Graph Sample and Aggregate (GraphSAGE) was also introduced for inductive representation learning, defined as

$$\boldsymbol{h_u^*} = \max_{v \in \mathcal{N}(u)} \boldsymbol{W h_v} + \boldsymbol{b}, \tag{2}$$

where $\boldsymbol{W}$ and $\boldsymbol{b}$ are a learnable linear transformation and bias term, respectively. GraphSAGE demonstrated strong performance on tasks requiring generalization to new and unseen graphs during evaluation. More recently, the Graph Isomorphism Network (GIN) was introduced as a maximally expressive GNN architecture for detecting graph isomorphism, rooted in the Weisfeiler-Lehman (1-WL) test (Weisfeiler & Leman, 1968), defined as

$$\boldsymbol{h_u^*} = \mathrm{MLP}\left((1+\varepsilon)\cdot\boldsymbol{h_u} + \sum_{v\in\mathcal{N}(u)}\boldsymbol{h_v}\right),\tag{3}$$

where MLP is a learnable multi-layer perceptron (MLP) and $\varepsilon$ is a learnable scalar parameter (Xu et al., 2019). GIN was shown to outperform other models in tasks where determining graph isomorphism becomes critical (Sato et al., 2021). Due to their simplicity and computational efficiency, GCN, GraphSAGE, and GIN became widely adopted across various applications (Li et al., 2020; Liu et al., 2022; Kim & Ye, 2020; Dwivedi et al., 2023; Hu et al., 2020). Notably, they may be classified as convolutional variants of GNN, as shown in Fig. 1(a), which may be expressed as

$$\bigoplus_{v\in\mathcal{N}(u)} c_{u,v}\cdot\psi\left(\boldsymbol{h_v}\right),\tag{4}$$

for some neighborhood aggregator $\bigoplus$ (*e.g.*, sum, mean, symmetric mean, and max). With this variant, messages from a neighboring node $v\in\mathcal{N}(u)$ to node $u$ are simply a function of the features of the neighboring node $\psi\left(\boldsymbol{h_v}\right)$ multiplied by a scalar factor $c_{u,v}$ based on the local graph structure.

## 2.2 Attentional Variant

Following the introduction of the Transformer, researchers considered incorporating the attention mechanism into the graph convolution operation to boost model performance. One of the earliest works was the Graph Attention Network (GAT), defined as

$$\boldsymbol{h_u^*} = \sum_{v\in\mathcal{N}(u)} \alpha_{u,v}\cdot\boldsymbol{Wh_v},\tag{5}$$

$$\alpha_{u,v} = \mathrm{Softmax}\left(e_{u,v}\right),\tag{6}$$

$$e_{u,v} = \mathrm{LeakyReLU}\left(\boldsymbol{a}^\top\left(\boldsymbol{Wh_u} + \boldsymbol{Wh_v}\right)\right),\tag{7}$$

where $\boldsymbol{W}$ and $\boldsymbol{a}$ are learnable linear transformations (Veličković et al., 2018). Other works, such as GATv2 (Brody et al., 2022), build upon GAT by proposing different methods for computing the attention scores $e_{u,v}$ for various applications (Wang et al., 2021; Hsu et al., 2023; Jiang et al., 2023). These GNN architectures may then be aptly classified as attentional GNN variants, as shown in Fig. 1(b), conventionally expressed as

$$\bigoplus_{v\in\mathcal{N}(u)} \alpha\left(\boldsymbol{h_u}, \boldsymbol{h_v}\right)\cdot\psi\left(\boldsymbol{h_v}\right).\tag{8}$$

In this variant, messages from a neighboring node $v\in\mathcal{N}(u)$ to node $u$ are still a function of the features of the neighboring node $\psi\left(\boldsymbol{h_v}\right)$. However, the scalar factor now becomes a function of both node features $\alpha\left(\boldsymbol{h_u}, \boldsymbol{h_v}\right)$, allowing it to dynamically adjust the contribution of each message based on the relative importance of the neighboring node.

It is also worth noting that practical applications of the attentional variant almost exclusively employ multiple attention heads to expand the representational capacity. By extracting multiple, independent representations in parallel and subsequently concatenating or averaging them, the *multi-head* attentional variant may capture diverse localized representations (Veličković et al., 2018). While this mechanism boosts the expressivity of the original attentional variant, it fundamentally remains an aggregation of messages calculated in a pair-wise manner. Crucially, the attentional variant, even with the dynamic attention of GATv2, is still only designed to consider the entire local neighborhood representation for post-message-calculation dynamic weighting during the aggregation step.

### 2.3 Message-Passing Variant

The message-passing variant provides a more general framework for the graph convolution operation in GNNs, leading to many architectures tailored for specific applications (Lim et al., 2025b;c). A prominent example is the Message-Passing Neural Network (MPNN), which was shown to perform well in approximating quantum mechanical simulations, even achieving an order-of-magnitude decrease in computational time (Gilmer et al., 2017). More recently, the Soft-Isomorphic Relational Graph Convolution Network (SIR-GCN) was introduced as a simple and computationally efficient architecture with maximal graph representational power within the bounds of localized message-passing, defined as

$$h_u^* = \sum_{v \in \mathcal{N}(u)} W_R \, \sigma \left( W_Q h_u + W_K h_v \right), \tag{9}$$

where $\sigma$ is a non-linear activation function and $W_R$, $W_Q$, $W_K$ are learnable linear transformations (Lim et al., 2025a). Owing to its message-passing *flavor*, SIR-GCN was even shown to mathematically generalize GCN, GraphSAGE, GIN, and GAT, among others. GNN architectures following these designs may be classified as message-passing variants, as shown in Fig. 1(c), expressed as

$$\bigoplus_{v \in \mathcal{N}(u)} \psi \left( h_u, h_v \right). \tag{10}$$

Crucially, messages from a neighboring node $v \in \mathcal{N}(u)$ to node $u$ in this variant are now a (potentially non-linear) function of both node features $\psi \left( h_u, h_v \right)$. This design makes it highly expressive, as it may be able to learn complex relationships between neighboring nodes beyond simple scalars.

### 2.4 Beyond One-Hop Localized Message-Passing

While one-hop localized GNN architectures are widely adopted due to their simplicity and scalability, their expressivity is strictly upper bounded by the 1-WL test (Xu et al., 2019). To overcome this fundamental limitation, several distinct paradigms have emerged that either abandon or heavily augment the standard message-passing framework.

#### 2.4.1 Multiple Aggregators & Higher-Order Neighborhoods

One immediate extension involves utilizing multiple aggregators (*e.g.*, sum, max, min, and standard deviation) to capture higher-order statistical moments of the neighborhood feature distribution, as seen with the Principal Neighborhood Aggregation (PNA) (Corso et al., 2020) and Efficient Graph Convolution (EGC-S, EGC-M) (Tailor et al., 2022). Beyond multiple aggregators, architectures such as $k$-dimensional GNNs ($k$-GNNs) (Morris et al., 2019), Folklore GNNs (FGNN) (Azizian & Lelarge, 2021), and Cellular Weisfeiler-Lehman Networks (CWNs) (Bodnar et al., 2021) process higher-order neighborhoods or cellular complexes to capture localized topological properties that go beyond the 1-WL test.

#### 2.4.2 Subgraph-Enhanced GNNs & Biconnectivity

Recognizing that non-isomorphic graphs often contain easily distinguishable subgraphs, architectures like the Equivariant Subgraph Aggregation Networks (ESAN) (Bevilacqua et al., 2022) and GNN As Kernel (GNN-AK) (Zhao et al., 2022) represent single graphs as bags of subgraphs (*e.g.*, node-deleted multisets or $k$-hop ego-networks). By applying equivariant processing to these extracted subgraphs, they establish provable theoretical upper bounds that strictly exceed the 1-WL test. Furthermore, recent advancements have shifted the focus toward graph biconnectivity as a superior expressivity metric. Models utilizing the Generalized Distance Weisfeiler-Lehman (GD-WL) framework are mathematically proven to resolve complex biconnectivity metrics, such as effective resistance, which classical message-passing operations fundamentally cannot compute (Zhang et al., 2023). Notably, the extraction and independent processing of ego-networks or biconnectivity metrics incur substantial, non-linear computational and memory overhead.

### 2.4.3   Graph Transformers & Global Context

To completely bypass the over-squashing and expressivity bottlenecks of localized subgraphs, several works have adapted the Transformer architecture for graph-structured data. Graphormer notably advances this by integrating structural priors through spatial distance, centrality, and edge encodings directly into the self-attention calculation (Ying et al., 2021). Moreover, the Spectral Attention Network (SAN) leverages the full Laplacian spectrum via learned positional encodings to provide a fully-connected, globally-aware attention mechanism (Kreuzer et al., 2021). Similarly, GraphGPS provides a highly scalable recipe that decouples localized real-edge message passing from fully-connected global attention, achieving linear complexity while serving as a universal function approximator (Rampášek et al., 2022). These models explicitly capture global inter-neighbor relationships through sophisticated structural and spectral encodings.

### 2.4.4   Pattern-Based & Graph Foundation Models

The most aggressive departure from the message-passing framework involves treating graph substructures as discrete, sequence-based tokens. The Neural Graph Pattern Machine (GPM) (Wang et al., 2025b) and the Generative Graph Pattern Machine (G2PM) (Wang et al., 2025d) utilize random walk tokenizers to extract task-relevant structural patterns, modeling them sequentially via Transformers to capture long-range dependencies without the inherent inductive biases of message-passing. In the pursuit of Graph Foundation Models (GFMs) (Wang et al., 2025a), frameworks like the Graph Foundation Model with Transferable Tree Vocabulary (GFT) (Wang et al., 2024) and the Graph Generality Identifier on Task-Trees (GIT) (Wang et al., 2025c) formulate the classical message-passing receptive field as discrete computation trees. These depth-1 task-trees serve as a universal, transferable vocabulary to align heterogeneous graph tasks across diverse domains. Finally, structure-free inference techniques, such as SimMLP, demonstrate that structural dependencies can be distilled directly into multi-layer perceptrons via self-supervised alignment, bypassing neighborhood fetching entirely during inference (Wang et al., 2025e).

### 2.4.5   Distinction with Current Work

While these advanced paradigms achieve state-of-the-art expressivity, they largely necessitate a fundamental departure from the localized message-passing inductive bias, often resulting in large computational and memory overheads. Fundamentally, the current work distinguishes itself with its straightforward and highly scalable architectural design for enhancing graph representations. Notably, multi-aggregator architectures (*e.g.*, PNA, EGC-M) implicitly approximate neighborhood context by concatenating independent, post-hoc statistical moments of the multiset of pair-wise messages. In contrast, the proposed NCMP framework explicitly constructs a unified, continuous representation of the multiset of neighborhood features prior to message calculation to directly inform the non-linear transformations of pair-wise feature interactions based on the aggregate distribution. Furthermore, Subgraph GNNs (*e.g.*, ESAN, GNN-AK) incur massive memory footprints by extracting overlapping ego-networks, and Graph Transformers (*e.g.*, Graphormer, SAN, GraphGPS) abandon local topologies for globally-connected attention. Meanwhile, SINC-GCN treats the immediate one-hop local neighborhood as a depth-1 computation tree, projecting this local task-tree into a continuous context vector and injecting it into the message calculation. Overall, this explicit architectural design of NCMP and SINC-GCN preserves the strict one-hop inductive bias and results in a highly efficient yet powerful localized message-passing framework.

## 3   A Framework for Neighborhood-Contextualized Message-Passing

To motivate the development of a new GNN framework, Table 1 first compares the three existing variants. In particular, previous work has shown how contextualized messages—messages that are sufficiently expressive functions, *i.e.*, universal function approximators, of the features of both the center node $\boldsymbol{h_u}$ and neighboring node $\boldsymbol{h_v}$—are crucial in boosting graph representational power (Lim et al., 2025a). Notably, both the convolutional and attentional variants do not possess this property as their core message $\psi$ solely considers the features of the neighboring node $\boldsymbol{h_v}$. Meanwhile, the message-passing variant *may* possess this property provided the message $\psi$ has universal function approximation capabilities.

Table 1: Comparison of Graph Neural Network Variants.

| GNN Variant | Contextualized Messages | Neighborhood-Contextualized |
|---|:---:|:---:|
| Convolutional | ✗ | ✗ |
| Attentional | ✗ | ✓ |
| Message-Passing | ✓ | ✗ |

In addition to this dimension, this work also highlights an implicit yet notable property of the attentional variant. Crucially, while it is typically expressed as Eq. (8), it is mathematically more accurate to express it as

$$\bigoplus_{v \in \mathcal{N}(u)} \alpha\left(\boldsymbol{h_u}, \boldsymbol{h_v}, \{\boldsymbol{h_w} : w \in \mathcal{N}(u)\}\right) \cdot \psi\left(\boldsymbol{h_v}\right), \tag{11}$$

to explicitly capture the dependency of the scalar attention weight $\alpha$ on the entire set of neighborhood features for normalization. Rooted in this key insight and foundational works in set-based aggregation methods (Zaheer et al., 2017; Qi et al., 2017), this work first adapts the concept of **neighborhood-contextualization** for GNNs as the functional dependence of the convolution operation on the *entire* set of neighborhood features $\{\boldsymbol{h_w} : w \in \mathcal{N}(u)\}$ as additional context of the *broader* local neighborhood of the center node $u$.

Interestingly, only the attentional variant implicitly possesses neighborhood-contextualization. However, this simply serves as a scalar softmax normalization factor for the attention weights, restricting its ability to fully capture the continuous structural distribution of the one-hop neighborhood, as noted in Lim et al. (2025a). Meanwhile, both the convolutional and message-passing variants remain indifferent or agnostic to the broader context of the local neighborhood. Critically, the key architectural limitation of the message-passing variant lies with its *pair-wise* messages $\psi\left(\boldsymbol{h_u}, \boldsymbol{h_v}\right)$ only considering the features of the center node $\boldsymbol{h_u}$ and each neighboring node $\boldsymbol{h_v}$ for $v \in \mathcal{N}(u)$ *individually*. This design makes $\psi$ neighborhood-agnostic, limiting its ability to perform more complex reasoning on the relationship among the entire set of one-hop neighboring nodes $\mathcal{N}(u)$ and potentially limiting its expressivity.

To address the limitations of both the message-passing and attentional variants, this work integrates both contextualized messages and neighborhood-contextualization within GNNs to propose the **neighborhood-contextualized message-passing (NCMP)** framework, as shown in Fig. 2, expressed as

$$\bigoplus_{v \in \mathcal{N}(u)} \psi\left(\boldsymbol{h_u}, \boldsymbol{h_v}, \{\boldsymbol{h_w} : w \in \mathcal{N}(u)\}\right). \tag{12}$$

Notably, unlike the attentional variant, where the neighborhood-contextualization is solely in the scalar $\alpha$, NCMP extends this to the *multi-dimensional* $\psi$, adapting the *vector* of messages themselves based on the *entire* set of one-hop neighborhood features $\{\boldsymbol{h_w} : w \in \mathcal{N}(u)\}$ thereby equipping it with the ability to *learn more meaningful relationships within the local neighborhood*, which is computationally expensive or indirect to achieve with existing GNN variants. Intuitively, rather than asking "given the context of my neighbors, *how much information* should I send?" as with the attentional variant, the proposed framework asks "given the context of my neighbors, *what is the appropriate information* I should send?". Furthermore, it is also easy to see that NCMP generalizes the message-passing variant. Hence, the proposed framework is functionally more expressive than classical neighborhood-agnostic message-passing GNNs, as the former encapsulates a broader class of graph-based models. Specifically, rather than restricting to rigid, localized pairs, it allows for feature transformations to adapt based on the broader structural and semantic context.

## 3.1 Soft-Isomorphic Neighborhood-Contextualized Graph Convolution Network: A Conceptual Proof

Foundational works in order-invariant machine learning, such as Deep Sets (Zaheer et al., 2017) and PointNet (Qi et al., 2017), established that universal approximators over unordered multisets require permutation-invariant sum or max decompositions. These architectures were originally designed to process independent point clouds or isolated sets by computing a global symmetric representation over the entire input. Furthermore, they demonstrate the theoretical necessity of conditioning neural objective functions on the entire

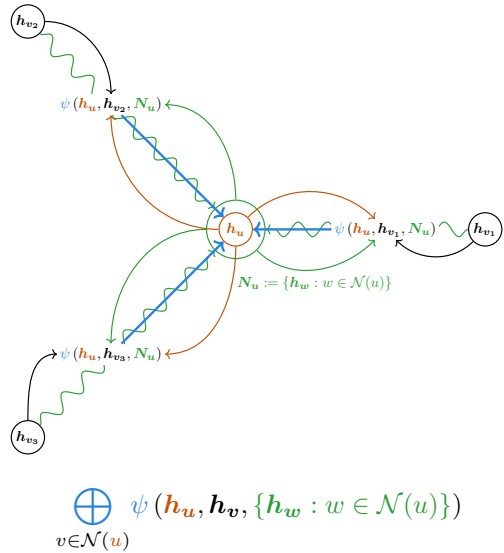

$$\bigoplus_{v \in \mathcal{N}(u)} \psi\left(\boldsymbol{h_u}, \boldsymbol{h_v}, \{\boldsymbol{h_w} : w \in \mathcal{N}(u)\}\right)$$

Figure 2: Neighborhood-Contextualized Message-Passing.

multiset to capture holistic context and maintain permutation invariance. While these set-based aggregation methods already treat their inputs as unordered sets and condition transformations on the full set, they generally operate as global readout functions for isolated point clouds.

In the current context of GNNs, one may adapt this set-based paradigm strictly to graph topologies by restricting the analysis to the local one-hop neighborhood features. In particular, while Eq. (12) provides an expressive, theoretically-grounded framework for enhancing localized message-passing architectures, it nevertheless still requires careful design choices. Critically, any operation on $\{\boldsymbol{h_w} : w \in \mathcal{N}(u)\}$ in an NCMP instance must be permutation-invariant, *i.e.*, strictly order-independent, while remaining flexible to arbitrary neighborhood size. One simple, practical, and mathematically grounded method for operationalizing NCMP is presented below.

By construction, since the message $\psi$ in Eq. (12) is a function of $\{\boldsymbol{h_w} : w \in \mathcal{N}(u)\}$, it is inherently neighborhood-contextualized. Moreover, following the theoretical development of SIR-GCN (Lim et al., 2025a), $\psi$ may be modeled as a two-layer MLP, guaranteeing contextualized messages. Leveraging the sum-decomposition theorems for continuous multiset functions established by Deep Sets (Zaheer et al., 2017) and PointNet (Qi et al., 2017), consider treating the neighborhood features as an identifiable, dynamic multiset and utilizing sum-decomposition or symmetric max-pooling operations to extract global representations. Using block matrix operations on the concatenated inputs $\boldsymbol{h_u}$, $\boldsymbol{h_v}$, and $\boldsymbol{h_w}$, one may then *initially* consider the equivalent parametrization

$$\boldsymbol{h_u^*} = \bigoplus_{v \in \mathcal{N}(u)} \boldsymbol{W_R}\, \sigma\left(\boldsymbol{W_Q h_u} + \boldsymbol{W_K h_v} + \boldsymbol{N_u}\right), \tag{13}$$

with the neighborhood context vector $\boldsymbol{N_u}$ defined as

$$\boldsymbol{N_u} \coloneqq \sum_{w \in \mathcal{N}(u)} \boldsymbol{W_N^{(w)} h_w}, \tag{14}$$

where $\bigoplus$ is some permutation-invariant aggregator (mimicking the symmetric pooling operations), $\sigma$ is a non-linear activation function, and $\boldsymbol{W_R}$, $\boldsymbol{W_Q}$, $\boldsymbol{W_K}$, $\boldsymbol{W_N^{(w)}}$ are learnable linear transformations.

In this specific formulation, $\boldsymbol{N_u}$ acts as a compressed, fixed-dimensional *vector* representation for the one-hop neighborhood features. It structurally mirrors the latent transformation $\sum \phi(\cdot)$ within the universal approximation theorems for continuous multiset functions detailed in Zaheer et al. (2017), while the subsequent

MLP acts as $\rho$. Furthermore, from the perspective of the more recent graph foundation models, the multiset $\{\boldsymbol{h_w} : w \in \mathcal{N}(u)\}$ may be interpreted as a continuous depth-1 task-tree or computation tree (Wang et al., 2024; 2025c). By projecting this depth-1 computation tree into the latent vector representation $\boldsymbol{N_u}$, SINC-GCN efficiently aligns local structural semantics, generalizing the analogous scalar normalization factor in standard softmax attention.

Crucially, however, this naive approach requires learning a distinct $\boldsymbol{W_N^{(w)}}$ for every node $w \in \mathcal{V}$. This is not merely *parameter inefficient* and *infeasible for inductive learning tasks*, including cold-start scenarios where purely structural MLPs like SimMLP excel (Wang et al., 2025e); it also fundamentally *violates the core symmetry constraints of GNNs.* Specifically, learning distinct $\boldsymbol{W_N^{(w)}}$ would tightly couple the model to an arbitrary node ordering, thereby breaking permutation equivariance.

To address these critical limitations, consider instead a constant $\boldsymbol{W_N}$ shared across all nodes $w \in \mathcal{V}$. This formulation not only promotes *parameter efficiency*, but is also a strict theoretical necessity to preserve both *permutation equivariance* and *generalizability.* Furthermore, other GNN aggregators may also be used in place of the sum aggregator in Eq. (14), as it is noted to exhibit difficulty generalizing to unseen graphs (Veličković et al., 2020). This approach further promotes flexibility while still respecting the permutation-invariance on $\{\boldsymbol{h_w} : w \in \mathcal{N}(u)\}$. Combining these features then results in the proposed **Soft-Isomorphic Neighborhood-Contextualized Graph Convolution Network (SINC-GCN)**[1] instantiation of the NCMP framework, expressed as

$$\boldsymbol{h_u^*} = \bigoplus_{v \in \mathcal{N}(u)} \boldsymbol{W_R} \, \sigma \left( \boldsymbol{W_Q h_u} + \boldsymbol{W_K h_v} + \bigotimes_{w \in \mathcal{N}(u)} \boldsymbol{W_N h_w} \right), \tag{15}$$

where $\bigoplus$ and $\bigotimes$ are some, potentially distinct, permutation-invariant aggregators (*e.g.*, sum, mean, symmetric mean, and max), $\sigma$ is a non-linear activation function, $\boldsymbol{W_R} \in \mathbb{R}^{d_\text{out} \times d_\text{hidden}}$, and $\boldsymbol{W_Q}, \boldsymbol{W_K}, \boldsymbol{W_N} \in \mathbb{R}^{d_\text{hidden} \times d_\text{in}}$. Moreover, for commutative aggregators $\bigoplus$ (*e.g.*, sum, mean, and symmetric mean), SINC-GCN has a computational complexity of

$$\mathcal{O}\left(|\mathcal{V}| \times d_\text{hidden} \times d_\text{in} + |\mathcal{E}| \times d_\text{hidden} + |\mathcal{V}| \times d_\text{out} \times d_\text{hidden}\right), \tag{16}$$

by leveraging linearity in Eq. (15), applying only an activation function along edges, and performing a two-step convolution constrained to the one-hop neighborhood receptive field. Specifically, extracting the neighborhood context representation only incurs an additional $\mathcal{O}\left(|\mathcal{V}| \times d_\text{hidden} \times d_\text{in}\right)$ overhead, making SINC-GCN comparable to classical one-hop localized GNNs in terms of asymptotic runtime complexity (Lim et al., 2025a), thereby underscoring its efficiency. Furthermore, SIR-GCN may also be viewed as an instance of SINC-GCN where $\boldsymbol{W_N} = \boldsymbol{0}$, positioning SIR-GCN as an *ablated model* of SINC-GCN.

Overall, SINC-GCN is a simple yet flexible *conceptual proof* of the proposed NCMP framework, grounded in established theoretical results for designing GNNs. By integrating both contextualized messages and neighborhood-contextualization, the proposed GNN architecture extends and generalizes classical one-hop localized GNNs while maintaining their computational efficiency.

### 3.1.1 Theoretical Bounds

As SIR-GCN was shown to be comparable to a modified 1-WL test (Lim et al., 2025a), it follows that SINC-GCN, as a generalization restricted to the one-hop neighborhood, retains the theoretical upper bounds of the 1-WL test and cannot formally distinguish graph families based on complex discrete isomorphisms or biconnectivity (Zhang et al., 2023) without additional positional or structural encodings.

Nonetheless, by integrating the aggregated neighborhood context representation $\boldsymbol{N_u}$ into the pair-wise message computation, SINC-GCN can capture localized structural semantics that standard message-passing fundamentally ignore. In particular, with standard message-passing, messages $\boldsymbol{\psi_{u,v}}$ from node $v$ to $u$ are strictly independent of the features of any other neighbor $\boldsymbol{h_w}$, $w \in \mathcal{N}(u) \backslash \{v\}$. Consequently, any modulation or adjustment to $\boldsymbol{\psi_{u,v}}$ based on the neighborhood feature distribution may only occur in the subsequent

---

[1]Lim et al. (2025a) defines the term soft-isomorphic.

layer, after the set of messages has been irreversibly compressed into a single vector representation, thereby exacerbating the over-squashing phenomenon. Conversely, by directly injecting $\boldsymbol{N_u}$ into the message calculation, SINC-GCN permits conditional message transformation based on the neighborhood feature distribution, with a non-zero, full-rank Jacobian

$$\frac{\partial}{\partial \boldsymbol{h_w}} \boldsymbol{\psi_{u,v}} \neq 0, \qquad \text{for } w \in \mathcal{N}(u) \setminus \{v\}. \tag{17}$$

Consequently, any change in $\boldsymbol{h_w}$ can fundamentally alter the message $\boldsymbol{\psi_{u,v}}$ along *any* feature space direction. This allows the model to learn continuous, local neighborhood properties—such as augmenting messages from outlier nodes if their features deviate significantly from the neighborhood norm captured in $\boldsymbol{N_u}$. Hence, the class of graph properties uniquely distinguishable by NCMP and the SINC-GCN instance involves continuous, relational feature dynamics and structural gating within the local one-hop neighborhood.[2]

## 4    Results

To demonstrate the practical utility of the proposed NCMP framework and the expressivity of SINC-GCN, this section provides an extensive analysis of its performance across both synthetic and benchmark datasets in node and graph property prediction tasks. Crucially, as the primary contribution of this work is the NCMP framework, the proposed SINC-GCN simply serves as an *illustrative instance* and is *not* explicitly designed to achieve state-of-the-art performance. Hence, only one-hop localized GNN architectures are used as baselines, ensuring a fair performance evaluation.

### 4.1    Synthetic Datasets

**UniqueSignature.** This original synthetic dataset consists of randomly generated graphs, each having 30 to 70 nodes with an edge creation probability $p_{\text{edge}}$ following the Erdős-Rényi model. Each node $u \in \mathcal{V}$ is also assigned an integer weight $w_u$ with $-W \leq w_u \leq W$. The task is then to identify *catalyst* nodes—nodes $u$ with a neighboring node $v \in \mathcal{N}(u)$ whose weight matches the total weight of all neighboring nodes of $u$, *i.e.*, $w_v = \sum_{w \in \mathcal{N}(u)} w_w$. As the graphs are generated randomly, the number of catalyst nodes naturally varies based on the generation configurations, as indicated by $\%_{\text{pos}}$. Motivated by previous works (Brody et al., 2022; Lim et al., 2025a), this dataset is *not* intended to serve as a comprehensive measure of general expressivity, but rather as a highly targeted *diagnostic probe* with an inherent structural bias, specifically and intentionally designed to highlight a specific limitation—neighborhood-agnosticism—inherent in existing message-passing architectures. By requiring the model to reason about the aggregate state of the local neighborhood, this diagnostic binary node classification problem empirically illustrates the necessity and utility of neighborhood-contextualization.

Table 2 presents the mean and standard deviation of the test balanced accuracy for SINC-GCN and baseline models—GCN, GraphSAGE, GATv2, GIN, and SIR-GCN—across different dataset configurations $W$ and $p_{\text{edge}}$ with varying percentage of positive class $\%_{\text{pos}}$. The performance for more advanced models—PNA, EGC-S, and EGC-M—is also presented as additional baselines. Notably, SINC-GCN consistently achieves perfect accuracy, attributed to its contextualized messages and neighborhood-contextualization. This design allows it to correctly identify *catalyst* nodes using the features of each neighboring node, contextualized on the entire set of neighborhood features. In fact, it may even be shown that with the appropriate choice of parameters $\bigoplus = \sum$, $\bigotimes = \sum$, $\sigma = \text{ReLU}$, $\boldsymbol{W_R} = [-1, -1]$, $\boldsymbol{W_Q} = \boldsymbol{0}$, $\boldsymbol{W_K} = [1, -1]^\top$, and $\boldsymbol{W_N} = [-1, 1]^\top$, SINC-GCN will mathematically always produce the correct classifications. Meanwhile, GCN, GraphSAGE, SIR-GCN, and EGC-S exhibit near-random performance, since their architectural design does not explicitly

---

[2]For attention head $h$ in multi-head GAT, $\boldsymbol{\psi_{u,v}^{(h)}} \coloneqq \alpha_{u,v}^{(h)} \cdot \boldsymbol{W^{(h)}} \boldsymbol{h_v}$. Hence, for $w \in \mathcal{N}(u) \setminus \{v\}$,

$$\frac{\partial}{\partial \boldsymbol{h_w}} \boldsymbol{\psi_{u,v}^{(h)}} = \boldsymbol{W^{(h)}} \boldsymbol{h_v} \left( \nabla_{\boldsymbol{h_w}} \alpha_{u,v}^{(h)} \right)^\top,$$

which is strictly rank-1. Consequently, any change in $\boldsymbol{h_w}$ can only scale the magnitude of the message $\boldsymbol{\psi_{u,v}^{(h)}}$ along its original feature space direction, and not project it into new semantic directions. Thus, NCMP is strictly more expressive and subsumes the multi-head attentional variant.

allow them to learn the appropriate relationship needed for this simple task. Likewise, GATv2 and GIN perform better than random on simpler dataset configurations, but fail to generalize well as problem complexity increases. In contrast, the performance of PNA and EGC-M is substantially better than random, as their use of the more *exotic* standard deviation aggregator *implicitly* involves the mean of the neighborhood features as standardization. Nevertheless, their performance comes with greater computational costs, as presented in Table 6 in Appendix B.1. Overall, the results illustrate the limitations of classical message-passing GNNs, the utility of both contextualized messages and neighborhood-contextualization, and the expressivity and efficiency of SINC-GCN.

Table 2: Test Balanced Accuracy on UniqueSignature.

| Model | Dataset Configuration | | | | | | | | |
|---|---|---|---|---|---|---|---|---|---|
| | $W = 1$ $p_{\text{edge}} = 0.3$ $\%_{\text{pos}} = 0.37$ | $W = 1$ $p_{\text{edge}} = 0.5$ $\%_{\text{pos}} = 0.29$ | $W = 1$ $p_{\text{edge}} = 0.7$ $\%_{\text{pos}} = 0.25$ | $W = 2$ $p_{\text{edge}} = 0.3$ $\%_{\text{pos}} = 0.35$ | $W = 2$ $p_{\text{edge}} = 0.5$ $\%_{\text{pos}} = 0.28$ | $W = 2$ $p_{\text{edge}} = 0.7$ $\%_{\text{pos}} = 0.24$ | $W = 3$ $p_{\text{edge}} = 0.3$ $\%_{\text{pos}} = 0.32$ | $W = 3$ $p_{\text{edge}} = 0.5$ $\%_{\text{pos}} = 0.27$ | $W = 3$ $p_{\text{edge}} = 0.7$ $\%_{\text{pos}} = 0.23$ |
| GCN | $0.50 \pm 0.00$ | $0.54 \pm 0.13$ | $0.59 \pm 0.18$ | $0.54 \pm 0.12$ | $0.63 \pm 0.19$ | $0.59 \pm 0.17$ | $0.54 \pm 0.11$ | $0.67 \pm 0.20$ | $0.67 \pm 0.21$ |
| GraphSAGE | $0.50 \pm 0.00$ | $0.50 \pm 0.00$ | $0.50 \pm 0.00$ | $0.50 \pm 0.00$ | $0.50 \pm 0.00$ | $0.50 \pm 0.00$ | $0.50 \pm 0.00$ | $0.50 \pm 0.00$ | $0.50 \pm 0.00$ |
| GATv2 | $0.84 \pm 0.17$ | $0.81 \pm 0.20$ | $0.73 \pm 0.23$ | $0.62 \pm 0.19$ | $0.67 \pm 0.21$ | $0.54 \pm 0.13$ | $0.66 \pm 0.19$ | $0.64 \pm 0.19$ | $0.50 \pm 0.00$ |
| GIN | $0.84 \pm 0.00$ | $0.85 \pm 0.00$ | $0.86 \pm 0.00$ | $0.82 \pm 0.00$ | $0.84 \pm 0.00$ | $0.85 \pm 0.00$ | $0.80 \pm 0.00$ | $0.84 \pm 0.00$ | $0.84 \pm 0.00$ |
| SIR-GCN | $0.50 \pm 0.00$ | $0.50 \pm 0.00$ | $0.50 \pm 0.00$ | $0.50 \pm 0.00$ | $0.50 \pm 0.00$ | $0.50 \pm 0.00$ | $0.50 \pm 0.00$ | $0.50 \pm 0.00$ | $0.50 \pm 0.00$ |
| SINC-GCN | $1.00 \pm 0.00$ | $1.00 \pm 0.00$ | $1.00 \pm 0.00$ | $1.00 \pm 0.00$ | $1.00 \pm 0.00$ | $1.00 \pm 0.00$ | $1.00 \pm 0.01$ | $1.00 \pm 0.00$ | $1.00 \pm 0.00$ |
| PNA | $1.00 \pm 0.00$ | $1.00 \pm 0.00$ | $1.00 \pm 0.00$ | $0.99 \pm 0.00$ | $1.00 \pm 0.00$ | $1.00 \pm 0.00$ | $0.98 \pm 0.00$ | $1.00 \pm 0.00$ | $1.00 \pm 0.00$ |
| EGC-S | $0.50 \pm 0.00$ | $0.54 \pm 0.11$ | $0.50 \pm 0.00$ | $0.58 \pm 0.16$ | $0.54 \pm 0.13$ | $0.50 \pm 0.00$ | $0.50 \pm 0.01$ | $0.54 \pm 0.12$ | $0.50 \pm 0.00$ |
| EGC-M | $1.00 \pm 0.00$ | $1.00 \pm 0.00$ | $0.98 \pm 0.05$ | $0.97 \pm 0.06$ | $0.96 \pm 0.08$ | $0.96 \pm 0.08$ | $0.94 \pm 0.08$ | $0.93 \pm 0.10$ | $0.96 \pm 0.09$ |

**Note**: blue: best one-hop localized model.

**Exp & CExp (Abboud et al., 2021).** As additional assessment of expressivity beyond the customized UniqueSignature dataset, the Exp and CExp datasets are also considered (Abboud et al., 2021). These synthetic datasets are explicitly crafted to evaluate GNN expressivity beyond the 1-WL test. Concretely, Exp contains pairs of non-isomorphic graph instances, representing propositional formulas, that are 1-WL indistinguishable but 2-WL distinguishable. The objective of this graph classification problem is then to determine whether or not the encoded formula is satisfiable. Building on Exp, the CExp augments half of the graph pairs (Corrupt) to make them 1-WL distinguishable by adding a minimal number of edges, while retaining the other half of the graph pairs ($\overline{\text{Exp}}$). This modification makes the problem more difficult than learning either of the subsets in isolation. Abboud et al. (2021) provides more details regarding the specific datasets.

Table 3: Test Accuracy on Exp & CExp.

| Model | Exp | Corrupt | $\overline{\text{Exp}}$ | CExp |
|---|---|---|---|---|
| GCN | $50.00 \pm 0.00$ | $64.50 \pm 3.17$ | $50.00 \pm 0.00$ | $57.25 \pm 1.58$ |
| GraphSAGE | $50.00 \pm 0.00$ | $50.00 \pm 0.00$ | $50.00 \pm 0.00$ | $50.00 \pm 0.00$ |
| GATv2 | $50.00 \pm 0.00$ | $61.17 \pm 12.47$ | $50.00 \pm 0.00$ | $55.58 \pm 6.24$ |
| GIN | $50.00 \pm 0.00$ | $53.33 \pm 7.92$ | $50.00 \pm 0.00$ | $51.67 \pm 3.96$ |
| SIR-GCN | $50.00 \pm 0.00$ | $51.83 \pm 5.50$ | $50.00 \pm 0.00$ | $50.92 \pm 2.75$ |
| SINC-GCN | $50.00 \pm 0.00$ | $85.50 \pm 20.62$ | $50.00 \pm 0.00$ | $67.75 \pm 10.31$ |
| PNA | $50.00 \pm 0.00$ | $99.83 \pm 0.50$ | $50.00 \pm 0.00$ | $74.92 \pm 0.25$ |
| EGC-S | $50.00 \pm 0.00$ | $50.00 \pm 0.00$ | $50.00 \pm 0.00$ | $50.00 \pm 0.00$ |
| EGC-M | $50.00 \pm 0.00$ | $65.83 \pm 22.06$ | $50.00 \pm 0.00$ | $57.92 \pm 11.03$ |

**Note**: blue: best one-hop localized model.

Table 3 presents the mean and standard deviation of the test accuracy for SINC-GCN, baseline models (GCN, GraphSAGE, GATv2, GIN, and SIR-GCN), and advanced models (PNA, EGC-S, and EGC-M) across both datasets. As expected, all models, including SINC-GCN, achieve 50% accuracy on the strictly

1-WL indistinguishable Exp and $\overline{\text{Exp}}$ instances, reflecting their theoretical upper bound. Crucially, however, the results on the 1-WL distinguishable Corrupt instances highlight the practical limitations of standard *pair-wise* message-passing. Despite being theoretically capable of distinguishing Corrupt instances, the baseline models exhibit poor performance. In contrast, SINC-GCN significantly outperforms them by a substantial margin, even achieving perfect accuracy in several seed initializations. This performance gap further demonstrates how neighborhood-contextualization provides a superior inductive bias for learning more general local neighborhood relationships, overcoming the limitations of standard message-passing. Furthermore, while PNA consistently solves Corrupt, it does so at significantly greater computational and memory overheads. Overall, the results underscore the practical utility of neighborhood-contextualization as a simple, highly scalable, and powerful inductive bias for localized message-passing.

## 4.2 Benchmark Datasets

**Benchmarking GNNs (Dwivedi et al., 2023).** This collection of benchmark datasets features a variety of mathematical and real-world graphs for various GNN tasks. Specifically, the WikiCS, PATTERN, and CLUSTER datasets are tailored for node property prediction tasks, while the MNIST, CIFAR10, and ZINC datasets are designed for graph property prediction tasks. Additionally, the mean absolute error (MAE) is the performance metric for ZINC, while accuracy is the primary metric for the remaining datasets. Collectively, these six datasets cover a diverse range of GNN applications, facilitating a comprehensive evaluation of model performance. Dwivedi et al. (2023) provides detailed information on the individual datasets.

**Open Graph Benchmark (Hu et al., 2020).** This collection of datasets offers realistic, extensive, and varied benchmarks suitable for GNNs. Specifically, the ogbn-arxiv dataset is used for node property prediction tasks. Meanwhile, the ogbg-molhiv dataset is designated for graph property prediction tasks. Accuracy serves as the performance metric for ogbn-arxiv, while the area under the receiver operating characteristic curve (ROC-AUC) is the primary metric for ogbg-molhiv. Hu et al. (2020) provides more details regarding the specific datasets.

Table 4: Test Performance on Benchmark Datasets.

| Model | WikiCS (↑) | PATTERN (↑) | CLUSTER (↑) | MNIST (↑) | CIFAR10 (↑) | ZINC (↓) | ogbn-arxiv (↑) | ogbg-molhiv (↑) |
|---|---|---|---|---|---|---|---|---|
| GCN | $77.47 \pm 0.85$ | $85.50 \pm 0.05$ | $47.83 \pm 1.51$ | $90.12 \pm 0.15$ | $54.14 \pm 0.39$ | $0.416 \pm 0.006$ | $71.92 \pm 0.21$ | $76.14 \pm 1.29$ |
| GraphSAGE | $74.77 \pm 0.95$ | $50.52 \pm 0.00$ | $50.45 \pm 0.15$ | $97.31 \pm 0.10$ | $65.77 \pm 0.31$ | $0.468 \pm 0.003$ | $71.73 \pm 0.26$ | $75.97 \pm 1.69$ |
| GATv2 | - | - | - | - | $67.48 \pm 0.53$ | $0.447 \pm 0.015$ | $71.87 \pm 0.43$ | $77.15 \pm 1.55$ |
| GIN | $75.86 \pm 0.58$ | $85.59 \pm 0.01$ | $58.38 \pm 0.24$ | $96.49 \pm 0.25$ | $55.26 \pm 1.53$ | $0.387 \pm 0.015$ | $67.33 \pm 1.47$ | $76.02 \pm 1.35$ |
| SIR-GCN | $78.06 \pm 0.66$ | $85.75 \pm 0.03$ | $63.35 \pm 0.19$ | $97.90 \pm 0.08$ | $71.98 \pm 0.40$ | $0.278 \pm 0.024$ | $72.52 \pm 0.16$ | $77.63 \pm 0.84$ |
| SINC-GCN | $78.17 \pm 0.68$ | $85.79 \pm 0.02$ | $63.51 \pm 0.15$ | $98.28 \pm 0.05$ | $73.37 \pm 0.41$ | $0.256 \pm 0.006$ | $72.66 \pm 0.09$ | $78.50 \pm 1.23$ |
| Cohen's $d$ | 0.1691 | 1.5104 | 0.8817 | 5.1610 | 3.4732 | 1.0801 | 0.9900 | 0.8894 |
| PNA | - | - | - | $97.19 \pm 0.08$ | $70.21 \pm 0.15$ | $0.320 \pm 0.032$ | $71.21 \pm 0.30$ | $79.05 \pm 1.32$ |
| EGC-S | - | - | - | - | $66.92 \pm 0.37$ | $0.364 \pm 0.020$ | $72.21 \pm 0.17$ | $77.44 \pm 1.08$ |
| EGC-M | - | - | - | - | $71.03 \pm 0.42$ | $0.281 \pm 0.007$ | $71.96 \pm 0.23$ | $78.18 \pm 1.53$ |
| CWN | - | - | - | - | - | $0.139 \pm 0.008$ | - | $80.55 \pm 1.04$ |
| GraphGPS | - | - | - | $98.05 \pm 0.13$ | $72.30 \pm 0.36$ | - | - | - |

**Notes**: blue: best one-hop localized model; **bold**: statistically significant by Welch's t-test at $\alpha = 0.05$ vs. best one-hop localized baseline model; Cohen's $d$: effect size vs best one-hop localized baseline model; missing values: no publicly published results.

Table 4 presents the mean and standard deviation of the test performance for SINC-GCN and baseline models—GCN, GraphSAGE, GATv2, GIN, and SIR-GCN—across the eight benchmark datasets. Crucially, the reported results for SINC-GCN follow the experimental configuration of Dwivedi et al. (2023) as presented in Appendix A, ensuring differences in model performance are primarily attributed to the GNN architecture. Notably, SINC-GCN largely achieves substantial performance gains beyond statistical significance with large to huge effect sizes, as measured by Cohen's $d$, all while operating with a *smaller hidden representation* and incurring only *minimal asymptotic computational overhead*. While the absolute performance gains on node property prediction tasks are modest, SINC-GCN achieves these with notably tighter standard deviations, highlighting how neighborhood-contextualization not only improves performance, but also makes the model more robust against random initialization variance. Crucially, the performance gains

are markedly more prominent for graph property prediction tasks. This disparity highlights a key advantage of the NCMP framework: neighborhood-contextualization is particularly vital for downstream graph-level representations while less critical for node-level representations, as empirically demonstrated in Appendix B.2. Because graph-level tasks require aggregating relational information across the entire network, equipping local message-passing with the broader structural and semantic context of the neighborhood allows the final readout function to capture a much richer global representation than neighborhood-agnostic baselines. The results thus reveal the significance of neighborhood-contextualization and position SINC-GCN as a performant and efficient alternative to classical GNNs.

Furthermore, the test performance for more advanced models—PNA, EGC-S, EGC-M, CWN, and GraphGPS—is also presented in Table 4 to provide a more comprehensive evaluation. While PNA and CWN achieve superior performance on ZINC and ogbg-molhiv, where preserving injectivity is critical, across the majority of datasets, these complex and computationally expensive GNN architectures fail to outperform the simpler and more efficient neighborhood-contextualization in SINC-GCN. Most notably, SINC-GCN achieves highly competitive or superior performance compared to GraphGPS on datasets like MNIST and CIFAR10. This demonstrates that effectively extracting a continuous depth-1 computation tree from the local neighborhood can offer a highly scalable, Pareto-efficient alternative to computationally expensive global attention mechanisms, achieving comparable expressivity without abandoning the sparse topological inductive bias.

### 4.2.1 Ablation & Sensitivity Analyses

To systematically validate the core contribution of the proposed NCMP framework, Table 5 further presents comprehensive ablation and sensitivity analyses across all benchmark datasets. The analyses evaluate SINC-GCN under various parameterizations of the neighborhood aggregator $\bigotimes$, including a direct ablation where $\bigotimes$ is entirely disabled (none).

Table 5: Ablation & Sensitivity Analysis on Benchmark Datasets.

| Aggregator $\bigotimes$ | WikiCS ($\uparrow$) | PATTERN ($\uparrow$) | CLUSTER ($\uparrow$) | MNIST ($\uparrow$) | CIFAR10 ($\uparrow$) | ZINC ($\downarrow$) | ogbn-arxiv ($\uparrow$) | ogbg-molhiv ($\uparrow$) |
|---|---|---|---|---|---|---|---|---|
| none | 78.06 ± 0.66 | 85.75 ± 0.03 | 63.35 ± 0.19 | 97.90 ± 0.08 | 71.98 ± 0.40 | 0.278 ± 0.024 | 72.52 ± 0.16 | 77.63 ± 0.84 |
| sum | FP Error | 85.74 ± 0.02 | 59.46 ± 0.09 | 97.97 ± 0.10 | 70.08 ± 1.25 | 0.258 ± 0.008 | 70.05 ± 0.74 | 77.33 ± 0.93 |
| max | 76.05 ± 1.69 | 85.64 ± 0.06 | 62.25 ± 0.32 | 98.13 ± 0.10 | 72.19 ± 0.60 | 0.262 ± 0.009 | 71.55 ± 0.16 | 75.22 ± 1.15 |
| mean | 78.17 ± 0.68 | 85.79 ± 0.02 | 63.51 ± 0.15 | 98.28 ± 0.05 | 73.37 ± 0.41 | 0.256 ± 0.006 | 72.22 ± 0.09 | 76.43 ± 0.77 |
| sym. mean | 78.00 ± 0.75 | 85.79 ± 0.02 | 63.38 ± 0.16 | 98.18 ± 0.11 | 73.16 ± 0.26 | 0.262 ± 0.017 | 72.66 ± 0.09 | 78.50 ± 1.23 |

**Ablating Neighborhood Context.** Setting $\bigotimes$ to none effectively eliminates neighborhood-contextualization from the convolution operation, directly reducing SINC-GCN to the baseline SIR-GCN. The results demonstrate a clear, consistent performance drop across almost all datasets. The performance gap is particularly prominent in tasks requiring deeper structural awareness. For instance, the MAE increases from 0.256 (mean) to 0.278 (none) in ZINC, while the accuracy drops from 73.37% (mean) to 71.98% (none) in CIFAR10. This direct ablation explicitly isolates the proposed neighborhood-contextualization and underscores the representational value of making the message-passing operation structurally and semantically aware of the broader local neighborhood.

**Sensitivity to Aggregator Choice.** Beyond the core ablation, Table 5 also illustrates the sensitivity of SINC-GCN to the specific $\bigotimes$ used to extract the neighborhood context representation. The sum aggregator exhibits severe instability—yielding a floating-point (FP) error on WikiCS—and generally sub-optimal performance across datasets like CLUSTER and ogbn-arxiv. This behavior highlights a critical sensitivity: unnormalized sum aggregations over highly variable neighborhood degrees lead to large feature variances and optimization failures (Veličković et al., 2020). Similarly, the max aggregator discards significant structural nuance by only retaining extreme feature values, resulting in less expressive neighborhood context representations compared to the optimal configurations (Xu et al., 2019). Conversely, the mean and symmetric mean (sym. mean) aggregators deliver the most robust and performant results across the eight datasets. By naturally normalizing the contextual vector against the neighborhood size, these aggregators provide a stable, smooth representation of the local structural and semantic context. Ultimately, this sensitivity

analysis validates the practical design of SINC-GCN: utilizing a simple, normalized representation of the neighborhood results in statistically significant performance gains over standard baselines, entirely avoiding the massive computational overhead of more advanced models.

Moreover, to ensure these performance gains are strictly attributed to neighborhood-contextualization and unconfounded by the number of parameters, Appendix B.3 further provides additional ablation and sensitivity analyses where SINC-GCN is evaluated against the SIR-GCN baseline at matched hidden representation dimensionality.

Overall, these benchmark datasets underscore the viability and potential of the proposed NCMP framework in offering a simple, practical, and efficient path toward enhancing the graph representational power of localized message-passing architectures via the integration of multiset neighborhood context within the message calculation.

## 5  Conclusion

In summary, the contribution of this work is threefold. It first adapts the concept of **neighborhood-contextualization** for GNNs, motivated by set-based aggregation methods and an implicit property of the attentional variant. It then proposes a practical generalization of the message-passing variant called **neighborhood-contextualized message passing (NCMP)**, which features both contextualized messages and neighborhood-contextualization. To illustrate its practical utility, a theoretically-grounded method for parametrizing NCMP is presented, leading to the development of the proposed **Soft-Isomorphic Neighborhood-Contextualized Graph Convolution Network (SINC-GCN)** as a pragmatic and scalable *conceptual proof* of the proposed framework. A comprehensive evaluation, spanning both synthetic and benchmark datasets in node and graph property prediction tasks, demonstrates how SINC-GCN achieves consistent and substantial gains against baseline GNN architectures, achieving a highly favorable Pareto-efficiency between representational capacity and computational overhead. Overall, the results underscore the potential of SINC-GCN for various GNN applications and the practical contribution of the proposed NCMP framework in enhancing the representational power of classical GNNs via local structural semantics.

Future works may consider applying SINC-GCN to critical domains where reasoning over localized structural semantics is paramount, such as detecting fraudulent behaviors in financial transaction networks or deciphering interaction fingerprints in protein-protein networks. Furthermore, more sophisticated NCMP parametrizations can be investigated, integrating advanced GNN paradigms like higher-order message-passing, graph kernels, and graph foundation models. Finally, the theoretical foundations of the paper open promising avenues for addressing the architectural limitations of modern large language models (LLMs) (Vaswani et al., 2017). In particular, the Transformer architecture may be interpreted from a geometric perspective as a softmax attentional GNN, where tokens act as nodes and attention weights represent learned, dynamic adjacency weights restricted by the directed topological constraint of the causal mask (Joshi, 2020). Consequently, modern LLMs remain susceptible to the rank-one Jacobian bottleneck discussed in Section 3.1.1, which exacerbates over-squashing and representational collapse (Barbero et al., 2024). Nonetheless, given the strong empirical performance of the proposed NCMP framework over neighborhood-agnostic attentional variants, better LLM architectures may be developed by replacing the standard softmax attention with an NCMP-parametrized architecture. This then treats the causally masked context window as a continuous multiset neighborhood context, allowing the LLM to capture richer structural semantics.

### Acknowledgments

This work is supported by Kyoto University and Toyota Motor Corporation through the joint project titled "Advanced Mathematical Science for Mobility Society."

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

# A  Experimental Set-up

The reported results for the synthetic dataset are obtained from the models at the final epoch across 5 seed initializations, while results for the benchmark datasets are obtained from the models with the best validation loss across 5 seed initializations. All experiments are conducted on a single NVIDIA® A800 (40GB) GPU using Deep Graph Library (DGL) with PyTorch backend. The codes to reproduce the results are published in the `SINC-GCN` repository.

## A.1  Synthetic Datasets

**UniqueSignature.** The models are trained using a set of 4,000 graphs and evaluated against a separate set of 1,000 graphs. These graphs are generated using the Erdős-Rényi model, each having 30 to 70 nodes with an edge creation probability $p_{\text{edge}}$. All reported results use a single GNN layer with 16 hidden units. Moreover, a two-layer MLP is used for GIN, while both PNA and EGC-M use the sum, max, and standard deviation aggregators. The models are then trained using the AdamW optimizer for 500 epochs with a $1 \times 10^{-3}$ learning rate and a batch size of 256. The learning rate is also scheduled to decay by a factor of 0.5 with a patience of 10 epochs based on the training loss.

**Exp & CExp (Abboud et al., 2021).** Following the experimental set-up of Abboud et al. (2021), the reported results use 8 GNN layers with 64 hidden units. Additionally, a two-layer MLP is used for GIN, while both PNA and EGC-M use the sum, max, and standard deviation aggregators. The models are then trained using the AdamW optimizer for 500 epochs with a $1 \times 10^{-3}$ learning rate and a batch size of 20. The learning rate is also scheduled to decay by a factor of 0.7 with a patience of 5 epochs based on the validation loss.

## A.2  Benchmark Datasets

**Benchmarking GNNs (Dwivedi et al., 2023).** Following the experimental set-up of Dwivedi et al. (2023); Corso et al. (2020); Tailor et al. (2022); Bodnar et al. (2021); Rampášek et al. (2022); Lim et al. (2025a), the reported results for SINC-GCN also use 4 GNN layers, employing batch normalization and residual connections, while constrained to a 100,000 parameter budget without extensive tuning. Consequently, SINC-GCN operates with a smaller hidden representation due to the additional parameters $\boldsymbol{W_N}$. To prevent overfitting, weight decays of rate $1 \times 10^{-1}$ and dropouts with rates in $\{0.1, 0.2, 0.3\}$ are also employed. Additionally, $\bigoplus$ is chosen as either the mean, symmetric mean, or max aggregator, similar to SIR-GCN (Lim et al., 2025a), while $\bigotimes$ is simply chosen as the mean aggregator. The graph readout function is chosen as the sum aggregator for ZINC and the mean aggregator for MNIST and CIFAR10. The models are then trained using the AdamW optimizer for a maximum of 500 epochs with a $1 \times 10^{-3}$ learning rate and a batch size of 128, when applicable. The learning rate is also scheduled to decay by a factor of 0.5 with a patience of 10 epochs based on the validation loss. The results for other models in Table 4 are obtained from previous works.

**Open Graph Benchmark (Hu et al., 2020).** Following the experimental set-up of Corso et al. (2020); Tailor et al. (2022); Lim et al. (2025a), the reported results for SINC-GCN also use 4 GNN layers, employing batch normalization and residual connections, while constrained to a 100,000 parameter budget without extensive tuning. Similarly, SINC-GCN operates with a smaller hidden representation due to the additional parameters $\boldsymbol{W_N}$. To prevent overfitting, weight decays with factors in $\{1 \times 10^{-3}, 1 \times 10^{-1}\}$ and dropouts with rates in $\{0.1, 0.2, 0.3, 0.4\}$ are also employed. Additionally, $\bigoplus$ is chosen as the mean aggregator, while $\bigotimes$ is simply chosen as the symmetric mean aggregator. The graph readout function is chosen as the mean aggregator for ogbg-molhiv. The models are then trained using the AdamW optimizer for a maximum of 1000 epochs with a learning rate in $\{1 \times 10^{-3}, 1 \times 10^{-2}\}$ and a batch size of 64 for ogbg-molhiv. The learning rate is also scheduled to decay by a factor of 0.5 with a patience of 10 or 50 epochs based on the validation loss. The results for other models in Table 4 are obtained from previous works.

# B  Additional Results

This section presents additional results to better position the contribution of the proposed NCMP framework and SINC-GCN.

## B.1  Runtime Analysis

The inference runtime for each model in UniqueSignature is presented in Table 6. These figures underscore how the proposed GNN architecture achieves a strong balance between model expressivity and computational efficiency. In particular, SINC-GCN has an inference runtime comparable to the baseline models—GCN, GraphSAGE, GATv2, GIN, and SIR-GCN—yet is functionally more expressive than these architectures. Moreover, when considered alongside the results in Table 2, they highlight PNA and EGC-M incurring significantly higher computational costs for their performance, in stark contrast to the significantly shorter runtime of SINC-GCN. Overall, these additional results demonstrate how SINC-GCN represents a balanced, Pareto-optimal middle ground—it retains continuous, local topological reasoning via the NCMP framework without incurring the massive computational overhead of higher-order or global attention architectures.

Table 6: UniqueSignature Inference Runtime.

| Model | Dataset Configuration | | | | | | | | |
|---|---|---|---|---|---|---|---|---|---|
| | $W = 1$ $p_{edge} = 0.3$ $\%_{pos} = 0.37$ | $W = 1$ $p_{edge} = 0.5$ $\%_{pos} = 0.29$ | $W = 1$ $p_{edge} = 0.7$ $\%_{pos} = 0.25$ | $W = 2$ $p_{edge} = 0.3$ $\%_{pos} = 0.35$ | $W = 2$ $p_{edge} = 0.5$ $\%_{pos} = 0.28$ | $W = 2$ $p_{edge} = 0.7$ $\%_{pos} = 0.24$ | $W = 3$ $p_{edge} = 0.3$ $\%_{pos} = 0.32$ | $W = 3$ $p_{edge} = 0.5$ $\%_{pos} = 0.27$ | $W = 3$ $p_{edge} = 0.7$ $\%_{pos} = 0.23$ |
| GCN | 0.32s ± 0.10s | 0.37s ± 0.09s | 0.41s ± 0.10s | 0.34s ± 0.12s | 0.38s ± 0.14s | 0.41s ± 0.10s | 0.34s ± 0.11s | 0.37s ± 0.08s | 0.40s ± 0.10s |
| GraphSAGE | 0.32s ± 0.10s | 0.37s ± 0.09s | 0.41s ± 0.10s | 0.33s ± 0.11s | 0.39s ± 0.14s | 0.41s ± 0.10s | 0.34s ± 0.11s | 0.39s ± 0.13s | 0.40s ± 0.10s |
| GATv2 | 0.33s ± 0.11s | 0.39s ± 0.13s | 0.41s ± 0.10s | 0.34s ± 0.11s | 0.39s ± 0.13s | 0.41s ± 0.10s | 0.34s ± 0.12s | 0.40s ± 0.13s | 0.41s ± 0.10s |
| GIN | 0.32s ± 0.10s | 0.39s ± 0.14s | 0.40s ± 0.10s | 0.34s ± 0.11s | 0.38s ± 0.13s | 0.40s ± 0.10s | 0.32s ± 0.10s | 0.37s ± 0.08s | 0.40s ± 0.10s |
| SIR-GCN | 0.34s ± 0.12s | 0.39s ± 0.13s | 0.41s ± 0.10s | 0.34s ± 0.12s | 0.38s ± 0.13s | 0.41s ± 0.10s | 0.34s ± 0.11s | 0.39s ± 0.14s | 0.40s ± 0.10s |
| PNA | 0.62s ± 0.10s | 0.72s ± 0.14s | 0.75s ± 0.10s | 0.64s ± 0.11s | 0.72s ± 0.14s | 0.75s ± 0.10s | 0.65s ± 0.12s | 0.72s ± 0.13s | 0.75s ± 0.09s |
| EGC-S | 0.39s ± 0.10s | 0.50s ± 0.15s | 0.56s ± 0.10s | 0.39s ± 0.11s | 0.48s ± 0.09s | 0.55s ± 0.10s | 0.39s ± 0.11s | 0.50s ± 0.15s | 0.55s ± 0.10s |
| EGC-M | 0.65s ± 0.10s | 0.92s ± 0.15s | 1.12s ± 0.09s | 0.65s ± 0.10s | 0.92s ± 0.15s | 1.12s ± 0.09s | 0.66s ± 0.10s | 0.90s ± 0.08s | 1.12s ± 0.09s |
| SINC-GCN | 0.34s ± 0.11s | 0.39s ± 0.13s | 0.40s ± 0.09s | 0.34s ± 0.11s | 0.39s ± 0.13s | 0.40s ± 0.10s | 0.32s ± 0.11s | 0.39s ± 0.13s | 0.40s ± 0.09s |

## B.2  Structural Analyses on Node Property Prediction Datasets

To more rigorously evaluate the effect of neighborhood-contextualization for node property prediction tasks, Tables 7 to 9 present additional structural analyses of the test performance on WikiCS, PATTERN, and CLUSTER, respectively, based on node degree and local clustering coefficient. The results demonstrate that the performance gaps between SINC-GCN and the baseline SIR-GCN remain uniformly minimal across all structural segments. For instance, on the PATTERN dataset, SINC-GCN achieves accuracies of 72.21%, 79.54%, 83.00%, and 82.37% across the first to fourth node degree quartiles, which closely mirrors the baseline SIR-GCN performance of 72.23%, 79.60%, 83.03%, and 82.36%, respectively. A similar result may also be observed when stratifying by the local clustering coefficient quartiles. These results empirically demonstrate that standard one-hop localized GNNs already capture sufficient local structural semantics for node property prediction tasks. Conversely, graph property prediction tasks require aggregating node-level information across the entire graph, making localized neighborhood contexts critical for preserving crucial information for downstream tasks. Overall, this analysis further contextualizes the results in Table 4 and better positions the contribution of the work.

Table 7: Structural Analysis on WikiCS.

| Model | Node Degree | | | | Clustering Coefficient | | | |
|---|---|---|---|---|---|---|---|---|
| | Q1 | Q2 | Q3 | Q4 | Q1 | Q2 | Q3 | Q4 |
| SIR-GCN | $72.01 \pm 1.50$ | $78.28 \pm 0.98$ | $78.34 \pm 0.84$ | $84.82 \pm 0.95$ | $71.11 \pm 1.60$ | $78.31 \pm 0.75$ | $79.52 \pm 0.91$ | $84.30 \pm 0.83$ |
| SINC-GCN | $71.54 \pm 1.54$ | $78.42 \pm 0.93$ | $78.02 \pm 0.86$ | $84.61 \pm 1.01$ | $70.72 \pm 1.73$ | $78.24 \pm 0.66$ | $79.30 \pm 0.93$ | $84.10 \pm 0.86$ |

Table 8: Structural Analysis on PATTERN.

| Model | Node Degree | | | | Clustering Coefficient | | | |
|---|---|---|---|---|---|---|---|---|
| | Q1 | Q2 | Q3 | Q4 | Q1 | Q2 | Q3 | Q4 |
| SIR-GCN | $72.23 \pm 0.14$ | $79.60 \pm 0.12$ | $83.03 \pm 0.07$ | $82.36 \pm 0.08$ | $86.44 \pm 0.06$ | $86.53 \pm 0.05$ | $85.62 \pm 0.04$ | $80.63 \pm 0.07$ |
| SINC-GCN | $72.21 \pm 0.06$ | $79.54 \pm 0.05$ | $83.00 \pm 0.08$ | $82.37 \pm 0.07$ | $86.47 \pm 0.10$ | $86.54 \pm 0.04$ | $85.58 \pm 0.06$ | $80.61 \pm 0.08$ |

Table 9: Structural Analysis on CLUSTER.

| Model | Node Degree | | | | Clustering Coefficient | | | |
|---|---|---|---|---|---|---|---|---|
| | Q1 | Q2 | Q3 | Q4 | Q1 | Q2 | Q3 | Q4 |
| SIR-GCN | $55.54 \pm 0.15$ | $62.36 \pm 0.13$ | $66.69 \pm 0.27$ | $69.74 \pm 0.39$ | $53.59 \pm 0.26$ | $60.79 \pm 0.28$ | $65.92 \pm 0.22$ | $73.25 \pm 0.15$ |
| SINC-GCN | $55.77 \pm 0.10$ | $62.46 \pm 0.19$ | $66.71 \pm 0.15$ | $69.81 \pm 0.17$ | $53.81 \pm 0.23$ | $60.87 \pm 0.21$ | $65.93 \pm 0.20$ | $73.37 \pm 0.07$ |

## B.3 Ablation & Sensitivity Analyses on Benchmark Datasets

Table 10 presents additional ablation and sensitivity analyses for SINC-GCN on the benchmark datasets. Crucially, different from Table 5, the new results match the dimension of the hidden representation of SINC-GCN to that of the baseline SIR-GCN with $\otimes$ set to none, thereby cleanly isolating the effect of neighborhood-contextualization from representational capacity. Across several datasets, the results remain largely robust, with the contextualized models outperforming the baseline. For instance, the symmetric mean aggregator achieves 63.67% on CLUSTER and an MAE of 0.259 on ZINC, maintaining a clear advantage over the baseline performance of 63.35% and 0.278, respectively. However, as SINC-GCN explicitly introduces an additional weight $W_N$ to extract the neighborhood context, matching the dimension of the hidden representation strictly increases its overall parameter count, consequently resulting in worse performance in some datasets. For instance, the mean aggregator lowers the accuracy on ogbg-molhiv from 77.63% to 75.74%. This performance degradation may be reasonably attributed to overfitting—since the neighborhood context already provides a highly expressive inductive bias, matching the dimension of the hidden representation makes it highly susceptible to over-parameterization. Overall, the additional analyses further validate the expressivity of SINC-GCN with its neighborhood-contextualization, while also demonstrating its parameter efficiency.

Table 10: Ablation & Sensitivity Analysis on Benchmark Datasets.

| Aggregator $\otimes$ | WikiCS ($\uparrow$) | PATTERN ($\uparrow$) | CLUSTER ($\uparrow$) | MNIST ($\uparrow$) | CIFAR10 ($\uparrow$) | ZINC ($\downarrow$) | ogbn-arxiv ($\uparrow$) | ogbg-molhiv ($\uparrow$) |
|---|---|---|---|---|---|---|---|---|
| none | $78.06 \pm 0.66$ | $85.75 \pm 0.03$ | $63.35 \pm 0.19$ | $97.90 \pm 0.08$ | $71.98 \pm 0.40$ | $0.278 \pm 0.024$ | $72.52 \pm 0.16$ | $77.63 \pm 0.84$ |
| sum | FP Error | $85.72 \pm 0.02$ | $59.80 \pm 0.13$ | $97.85 \pm 0.09$ | $70.46 \pm 1.20$ | $0.267 \pm 0.010$ | $70.58 \pm 0.74$ | $76.40 \pm 1.36$ |
| max | $76.30 \pm 1.80$ | $85.63 \pm 0.08$ | $62.58 \pm 0.37$ | $98.08 \pm 0.10$ | $73.06 \pm 0.32$ | $0.249 \pm 0.006$ | $71.61 \pm 0.18$ | $75.69 \pm 0.80$ |
| mean | $78.15 \pm 0.63$ | $85.78 \pm 0.03$ | $63.48 \pm 0.17$ | $98.25 \pm 0.13$ | $73.67 \pm 0.47$ | $0.278 \pm 0.026$ | $72.25 \pm 0.12$ | $75.74 \pm 1.48$ |
| sym. mean | $78.15 \pm 0.65$ | $85.78 \pm 0.02$ | $63.67 \pm 0.19$ | $98.23 \pm 0.07$ | $73.55 \pm 0.64$ | $0.259 \pm 0.013$ | $72.49 \pm 0.08$ | $77.54 \pm 2.10$ |

