# OpenReview forum: "Enhancing Graph Representations with Neighborhood-Contextualized Message-Passing"
_TMLR — Accepted by TMLR_

### Review · Reviewer_muiW · 2026-04-17

**Summary Of Contributions:**

This paper introduces Neighborhood-Contextualized Message Passing (NCMP), a formulation that augments standard pairwise message passing by conditioning messages on the entire set of one-hop neighbor features. The authors instantiate NCMP as SINC-GCN, which injects a permutation-invariant neighborhood summary into the pairwise message function while preserving linear-time complexity in the number of edges. Empirically, SINC-GCN shows strong performance on a diagnostic synthetic task and delivers consistent, statistically significant improvements over classical one-hop baselines on several node- and graph-level benchmarks, with modest overhead.

**Audience:**

Yes

**Audience Explanation:**

People who work on GNN may be interested.

**Claims And Evidence:**

No

**Claims Explanation:**

While the paper presents a technically sound and efficient architecture in SINC-GCN, its theoretical novelty claims appear to be overstated in several key areas.

Main Concerns

1. The Core Concept is Not New

The notion of conditioning message computation on the entire neighborhood is not novel. Set-based methods like DeepSets (Zaheer et al., 2017) and PointNet have long treated neighborhoods as sets. More directly, k-GNNs, subgraph GNNs, and Graph Transformers such as Graphormer, SAN, and GPS already capture inter-neighbor relationships, often more expressively. The paper's claim to "formalize neighborhood-contextualization" largely amounts to rebranding a well-established idea.

2. The Dismissal of Prior Work is Evasive

The paper acknowledges higher-order and transformer-based methods in Section 2.4, but deflects by citing their computational cost. This argument is weakened by the fact that one-hop methods like PNA and EGC-M — which implicitly capture neighborhood context through multi-aggregation — already exist and are even shown to outperform SINC-GCN on certain tasks within the paper's own experiments.

3. The Distinction from GAT is Exaggerated

The paper argues that GAT's softmax normalization is merely a scalar and thus limited. However, multi-head GAT with nonlinear combinations already approximates much of what the proposed framework offers. The architectural difference is incremental rather than fundamental.

4. Experimental Design Favors the Proposed Method

The UniqueSignature synthetic dataset is custom-designed by the authors specifically to expose neighborhood-agnosticism — a limitation that SINC-GCN is purpose-built to address. While illustrative, such a dataset cannot serve as an unbiased measure of general expressivity.
Fair Assessment

The engineering contribution of SINC-GCN is genuine: it is a clean, parameter-efficient method for incorporating neighborhood context within the one-hop message-passing paradigm. However, the paper's framing as a foundational theoretical framework is an overclaim. A more accurate characterization of the contribution would be: a practical and efficient instantiation of neighborhood-aware message-passing under strict one-hop constraints, rather than the introduction of an entirely new conceptual framework.

**Requested Changes:**

1. Reframe the Novelty Claims

The paper's framing of neighborhood-contextualization as a fundamentally new concept is its weakest point. The authors should reposition the contribution more precisely. Rather than claiming to formalize an entirely new paradigm, the paper should explicitly acknowledge that the broader idea of conditioning on neighborhood context is well-established, and articulate more carefully how SINC-GCN differs from and improves upon existing one-hop approaches specifically. A more defensible claim would center on the practical efficiency and simplicity of the proposed parametrization, rather than conceptual originality.

2. Expand and Strengthen the Related Work Section

The current treatment of related work in Section 2.4 is insufficiently thorough. The authors should engage more directly with the following lines of work and clarify the precise relationship between NCMP and each:

- Set-based aggregation methods (e.g., DeepSets, PointNet), which already treat neighborhoods as unordered sets and condition transformations on the full set
- One-hop methods with multi-aggregation (e.g., PNA, EGC-M), which implicitly capture neighborhood context and are direct competitors within the same computational budget
- Graph Transformers (e.g., Graphormer, SAN, GPS), which explicitly model pairwise inter-neighbor relationships via attention
- Subgraph and higher-order GNNs, beyond the brief dismissal currently offered

Simply citing these works and attributing their limitations to computational cost is insufficient. The authors should provide a formal or empirical argument distinguishing NCMP from these approaches.

3. Address the GAT Comparison More Rigorously

The paper argues that GAT's neighborhood-contextualization is limited to a scalar softmax weight, and uses this to motivate NCMP. However, this comparison overlooks multi-head GAT and its nonlinear variants. The authors should either provide a formal proof that multi-head attentional variants cannot subsume NCMP, or temper their claims about the attentional variant's limitations accordingly.

4. Improve the Synthetic Dataset Design

The UniqueSignature dataset is constructed specifically to highlight neighborhood-agnosticism, which is precisely the limitation that SINC-GCN is designed to overcome. While such diagnostic datasets are useful, the authors should acknowledge this inherent bias explicitly. To strengthen the experimental validation, the authors should consider including additional synthetic benchmarks that are not designed around SINC-GCN's specific strengths, such as existing synthetic benchmarks from the GNN expressivity literature (e.g., EXP, CEXP, or SR25).

5. Provide Stronger Theoretical Characterization

The paper acknowledges that SINC-GCN remains bounded by the 1-WL test. However, it would strengthen the theoretical contribution significantly to provide a more precise characterization of exactly what SINC-GCN can express that standard message-passing cannot, beyond informal intuitions about structural semantics. Specifically, the authors should formalize the class of graph properties that become distinguishable under NCMP but not under classical message-passing, grounding the practical performance gains in a rigorous theoretical explanation.

6. Contextualize Benchmark Results More Carefully

On node property prediction tasks, the performance gains of SINC-GCN over baselines are modest. The authors attribute this to the lesser importance of neighborhood context in node-level tasks, which is a reasonable hypothesis, but it should be supported by more rigorous analysis. For instance, ablation studies isolating the effect of neighborhood-contextualization on different graph structural properties (e.g., degree heterogeneity, clustering coefficient) would make this argument more convincing.

7. Clarify the Parameter Budget Constraint

SINC-GCN operates with a smaller hidden representation than baselines under the same parameter budget due to the additional weights introduced by W_N​. This means part of the performance comparison may not be entirely controlled. The authors should include an additional experiment comparing SINC-GCN against baselines at equivalent hidden dimensionality rather than equivalent parameter count, to more cleanly isolate the effect of neighborhood-contextualization from representational capacity.

---

> ### Author Response · Authors · 2026-05-03
>
> We thank the reviewer for the constructive and thorough feedback. We have revised the manuscript to address the requested changes as follows:
>
> 1. **Softened Novelty Claims.** We have comprehensively revised the manuscript to temper and soften our claims regarding novelty and contributions. Specifically, we now explicitly state that neighborhood-contextualization adapts and refines well-established concepts from set-based aggregation literature for the specific context of GNNs. With this as basis, the proposed NCMP framework functions as a practical generalization of the standard message-passing variant. Finally, SINC-GCN is presented as a simple, mathematically grounded method for its efficient parametrization and operationalization.
>
> 2. **Strengthened Related Works.** We substantially expanded Section 2.4 to provide a more comprehensive discussion of related works. This section now includes dedicated subsections on: 1) Multiple Aggregators & Higher-Order Neighborhoods, 2) Subgraph-Enhanced GNNs & Biconnectivity, 3) Graph Transformers & Global Context, and 4) Pattern-Based & Graph Foundation Models. Furthermore, we added a dedicated subsection to formally articulate the architectural distinction between the NCMP framework and these advanced paradigms. Finally, a discussion of set-based aggregation methods was integrated into Section 3.1 to properly motivate and situate the formulation of SINC-GCN in relation to these established theories.
>
> 3. **Expanded Comparison with Attentional Variant.** In Section 2.2, we now explicitly acknowledge that practical applications almost exclusively employ multi-head attentional variants to capture diverse localized representations. Nevertheless, we highlight that even with non-linear and dynamic attention mechanisms like GATv2, this variant is still only designed to consider the entire local neighborhood representation for post-message-calculation dynamic weighting during the aggregation step. To formalize this distinction, Section 3.1.1 provides a formal mathematical analysis demonstrating that the NCMP framework is strictly more expressive than, and effectively subsumes, the multi-head attentional variant.
>
> 4. **Added Synthetic Datasets.** Within Section 4.1, we explicitly acknowledge the inherent structural bias of the UniqueSignature dataset, acknowledging its tailored design for SINC-GCN. To provide an additional assessment of expressivity beyond the customized dataset, we have incorporated the standard EXP and CEXP synthetic datasets. The empirical results confirm that SINC-GCN remains upper bounded by the 1-WL test. Crucially, on 1-WL distinguishable instances, SINC-GCN outperforms standard message-passing architectures by a considerable margin, even achieving perfect accuracy in several seed initializations. This underscores the practical utility of neighborhood-contextualization as a simple, highly scalable, and powerful inductive bias for localized message-passing.
>
> 5. **Improved Theoretical Discussion.** Section 3.1.1 has been substantially expanded to provide a more mathematically rigorous articulation of our theoretical contributions. We specifically characterize the class of graph properties uniquely distinguishable by NCMP and the SINC-GCN instance. This involves continuous, relational feature dynamics and structural gating within the local one-hop neighborhood, supporting the empirical results presented in Section 4.
>
> 6. **Contextualized Benchmark Results.** We have provided supplementary results in Appendix B.2 on structural analyses on node property prediction datasets, stratified by node degree and local clustering coefficient quartiles. These evaluations illustrate that the performance gaps between SINC-GCN and the baseline SIR-GCN remain uniformly minimal across all structural segments. This supports the hypothesis that standard one-hop localized GNNs already capture sufficient local structural semantics for node property prediction tasks.
>
> 7. **Added Ablation Results.** Appendix B.3 now includes additional ablation and sensitivity analyses that matches the dimension of the hidden representation of SINC-GCN to that of the baseline SIR-GCN with $\bigoplus$ set to none, thereby cleanly isolating the effect of neighborhood-contextualization from representational capacity. The new results remain largely robust, with the contextualized models outperforming the baseline. Nevertheless, as SINC-GCN explicitly introduces an additional weight $\boldsymbol{W_N}$ to extract the neighborhood context, increasing its overall parameters, and consequently resulting in worse performance in some datasets, which may be reasonably attributed to overfitting as the neighborhood context already provides a highly expressive inductive bias. Overall, the additional analyses further validate the expressivity of SINC-GCN with its neighborhood-contextualization, while also demonstrating its parameter efficiency.

---

> ### Author Response · Authors · 2026-05-03
> **Additional Results**
>
> > **Table: Test Accuracy on EXP & CEXP.**
>
> | **Model** | **$\text{EXP}$** | **$\text{Corrupt}$** | **$\overline{\text{EXP}}$** | **$\text{CEXP}$** |
> | :-------------- | :----------------------: | :--------------------------: | :---------------------------------: | :-----------------------: |
> | GCN             |      50.00 ± 0.00      |        64.50 ± 3.17        |            50.00 ± 0.00            |       57.25 ± 1.58       |
> | GraphSAGE       |      50.00 ± 0.00      |        50.00 ± 0.00        |            50.00 ± 0.00            |       50.00 ± 0.00       |
> | GATv2           |      50.00 ± 0.00      |        61.17 ± 12.47        |            50.00 ± 0.00            |       55.58 ± 6.24       |
> | GIN             |      50.00 ± 0.00      |        53.33 ± 7.92        |            50.00 ± 0.00            |       51.67 ± 3.96       |
> | SIR-GCN         |      50.00 ± 0.00      |        51.83 ± 5.50        |            50.00 ± 0.00            |       50.92 ± 2.75       |
> | SINC-GCN        |      50.00 ± 0.00      |        85.50 ± 20.62        |            50.00 ± 0.00            |      67.75 ± 10.31      |
> | PNA             |      50.00 ± 0.00      |        99.83 ± 0.50        |            50.00 ± 0.00            |       74.92 ± 0.25       |
> | EGC-S           |      50.00 ± 0.00      |        50.00 ± 0.00        |            50.00 ± 0.00            |       50.00 ± 0.00       |
> | EGC-M           |      50.00 ± 0.00      |        65.83 ± 22.06        |            50.00 ± 0.00            |      57.92 ± 11.03      |
>
> > **Table: Structural Analysis on WikiCS.**
>
> | **Model** | **Deg Q1** | **Deg Q2** | **Deg Q3** | **Deg Q4** | **CC Q1** | **CC Q2** | **CC Q3** | **CC Q4** |
> | :-------------- | :--------------: | :--------------: | :--------------: | :--------------: | :-------------: | :-------------: | :-------------: | :-------------: |
> | SIR-GCN         |  72.01 ± 1.50  |  78.28 ± 0.98  |  78.34 ± 0.84  |  84.82 ± 0.95  |  71.11 ± 1.60  |  78.31 ± 0.75  |  79.52 ± 0.91  |  84.30 ± 0.83  |
> | SINC-GCN        |  71.54 ± 1.54  |  78.42 ± 0.93  |  78.02 ± 0.86  |  84.61 ± 1.01  |  70.72 ± 1.73  |  78.24 ± 0.66  |  79.30 ± 0.93  |  84.10 ± 0.86  |
>
> > **Table: Structural Analysis on PATTERN.**
>
> | **Model** | **Deg Q1** | **Deg Q2** | **Deg Q3** | **Deg Q4** | **CC Q1** | **CC Q2** | **CC Q3** | **CC Q4** |
> | :-------------- | :--------------: | :--------------: | :--------------: | :--------------: | :-------------: | :-------------: | :-------------: | :-------------: |
> | SIR-GCN         |  72.23 ± 0.14  |  79.60 ± 0.12  |  83.03 ± 0.07  |  82.36 ± 0.08  |  86.44 ± 0.06  |  86.53 ± 0.05  |  85.62 ± 0.04  |  80.63 ± 0.07  |
> | SINC-GCN        |  72.21 ± 0.06  |  79.54 ± 0.05  |  83.00 ± 0.08  |  82.37 ± 0.07  |  86.47 ± 0.10  |  86.54 ± 0.04  |  85.58 ± 0.06  |  80.61 ± 0.08  |
>
> > **Table: Structural Analysis on CLUSTER.**
>
> | **Model** | **Deg Q1** | **Deg Q2** | **Deg Q3** | **Deg Q4** | **CC Q1** | **CC Q2** | **CC Q3** | **CC Q4** |
> | :-------------- | :--------------: | :--------------: | :--------------: | :--------------: | :-------------: | :-------------: | :-------------: | :-------------: |
> | SIR-GCN         |  55.54 ± 0.15  |  62.36 ± 0.13  |  66.69 ± 0.27  |  69.74 ± 0.39  |  53.59 ± 0.26  |  60.79 ± 0.28  |  65.92 ± 0.22  |  73.25 ± 0.15  |
> | SINC-GCN        |  55.77 ± 0.10  |  62.46 ± 0.19  |  66.71 ± 0.15  |  69.81 ± 0.17  |  53.81 ± 0.23  |  60.87 ± 0.21  |  65.93 ± 0.20  |  73.37 ± 0.07  |
>
> > **Table: Dimension-Matched Ablation & Sensitivity Analyses.**
>
> | **Aggregator ⨂** | **WikiCS (↑)** | **PATTERN (↑)** | **CLUSTER (↑)** | **MNIST (↑)** | **CIFAR10 (↑)** | **ZINC (↓)** | **ogbn-arxiv (↑)** | **ogbg-molhiv (↑)** |
> | :---------------------- | :-------------------: | :--------------------: | :--------------------: | :------------------: | :--------------------: | :-----------------: | :-----------------------: | :------------------------: |
> | none                    |     78.06 ± 0.66     |     85.75 ± 0.03     |     63.35 ± 0.19     |    97.90 ± 0.08    |     71.98 ± 0.40     |   0.278 ± 0.024   |       72.52 ± 0.16       |       77.63 ± 0.84       |
> | sum                     |       FP Error       |     85.72 ± 0.02     |     59.80 ± 0.13     |    97.85 ± 0.09    |     70.46 ± 1.20     |   0.267 ± 0.010   |       70.58 ± 0.74       |       76.40 ± 1.36       |
> | max                     |     76.30 ± 1.80     |     85.63 ± 0.08     |     62.58 ± 0.37     |    98.08 ± 0.10    |     73.06 ± 0.32     |   0.249 ± 0.006   |       71.61 ± 0.18       |       75.69 ± 0.80       |
> | mean                    |     78.15 ± 0.63     |     85.78 ± 0.03     |     63.48 ± 0.17     |    98.25 ± 0.13    |     73.67 ± 0.47     |   0.278 ± 0.026   |       72.25 ± 0.12       |       75.74 ± 1.48       |
> | sym. mean               |     78.15 ± 0.65     |     85.78 ± 0.02     |     63.67 ± 0.19     |    98.23 ± 0.07    |     73.55 ± 0.64     |   0.259 ± 0.013   |       72.49 ± 0.08       |       77.54 ± 2.10       |

---

> ### Author Response · Authors · 2026-05-03
> **Theoretical Discussion on Expressivity**
>
> > With standard (non-attentional) message-passing, messages $\boldsymbol{\psi_{u,v}}$ are strictly independent of $\boldsymbol{h_w}$, for $w \in \mathcal{N}(u) \setminus \{v\}$. Hence, any modulation to $\boldsymbol{\psi_{u,v}}$ based on the neighborhood feature distribution may only occur in the subsequent layer, after the set of messages has been irreversibly compressed into a single vector representation.
>
> > For attention head $h$ in multi-head GAT, $\boldsymbol{\psi_{u,v}^{(h)}} = \alpha_{u,v}^{(h)} \cdot \boldsymbol{W^{(h)}} \boldsymbol{h_v}$. Hence, for $w \in \mathcal{N}(u) \setminus \{v\}$, $\frac{\partial}{\partial \boldsymbol{h_w}} \boldsymbol{\psi_{u,v}^{(h)}} = \boldsymbol{W^{(h)}} \boldsymbol{h_v} \left(\nabla_{\boldsymbol{h_w}} \alpha_{u,v}^{(h)}\right)^\top$, which is strictly rank-1. Hence, any change in $\boldsymbol{h_w}$ can only **scale** $\boldsymbol{\psi_{u,v}^{(h)}}$ along its **original feature space direction**, and not project it into new semantic directions.
>
> > By directly injecting $\boldsymbol{N_u}$ into $\boldsymbol{\psi_{u,v}}$, SINC-GCN permits conditional message transformation based on the neighborhood feature distribution, with a non-zero, full-rank Jacobian $\frac{\partial}{\partial \boldsymbol{h_w}} \boldsymbol{\psi_{u,v}}$, for $w \in \mathcal{N}(u) \setminus \{v\}$. Consequently, any change in $\boldsymbol{h_w}$ can fundamentally alter $\boldsymbol{\psi_{u,v}}$ along **any feature space direction**. Hence, the class of graph properties uniquely distinguishable by NCMP and the SINC-GCN instance involves continuous, relational feature dynamics and structural gating within the local one-hop neighborhood.

---

### Review · Reviewer_5xAb · 2026-04-19

**Summary Of Contributions:**

The paper studies a limitation of the *message-passing* variant of one-hop GNNs: pairwise messages `ψ(h_u, h_v)` see only a single neighbor, while the *attentional* variant implicitly conditions on the whole neighborhood only through the softmax normalizer of a scalar attention weight. The authors (i) formalize a property they call **neighborhood-contextualization**, (ii) introduce a generalized operator family, **NCMP**, where messages take the form `ψ(h_u, h_v, {h_w : w ∈ N(u)})`, and (iii) instantiate this as **SINC-GCN** — SIR-GCN augmented with a shared linear context term `N_u = Σ_{w ∈ N(u)} W_N h_w`. SINC-GCN preserves permutation-equivariance, remains inductive, and adds an `O(|V| d_hidden d_in)` asymptotic cost. Empirically, (a) on a purpose-built synthetic *UniqueSignature* task, SINC-GCN achieves ~1.0 accuracy while classical one-hop baselines (GCN, GraphSAGE, GATv2, GIN, SIR-GCN, EGC-S) collapse to near-random; (b) on six Benchmarking-GNN datasets and two OGB datasets under a 100K-parameter budget, SINC-GCN reports statistically significant (Welch's t-test, α=0.05) gains over those baselines at comparable runtime; and (c) an ablation sweeping the context aggregator (sum / max / mean / symmetric-mean / none) supports the claim that neighborhood-contextualization drives the gain.

### Strengths

- **Clean conceptual framing.** The three-axis taxonomy (Table 1) and the rewrite of attentional aggregation as `α(h_u, h_v, {h_w}) · ψ(h_v)` in Eq. 11 together make the hidden neighborhood-dependence of the attentional variant explicit in a way I have not seen stated as crisply in prior work. NCMP (Eq. 12) is an honest super-set of both the attentional and the classical MP variants, and the paper is appropriately careful not to oversell (it acknowledges that NCMP does not lift the 1-WL bound and that SINC-GCN is a "conceptual proof").
- **Non-obvious equivariance argument.** The discussion on pp. 6–7 explaining why `W_N^{(w)}` must be tied across `w ∈ V` — otherwise permutation-equivariance breaks and inductive transfer is impossible — is a non-obvious architectural observation that well-motivates the specific form of SINC-GCN.
- **Rigorous empirical protocol.** UniqueSignature (Section 4.1) is a clean synthetic probe that cleanly separates SINC-GCN (1.00) from baselines (~0.5–0.85) and is not a single-configuration artefact (Table 2 sweeps W and p_edge). The benchmark protocol — 100K-parameter budget following the standard Benchmarking-GNN / OGB conventions, Welch's t-tests at α=0.05 instead of raw means, and explicit inference-time reporting (Table 5) — is more rigorous than many recent GNN papers. Table 4's `aggregator = none` ablation recovers SIR-GCN exactly.

### Weaknesses

- **Operational delta is smaller than the rhetorical framing, and the expressivity claim is rhetorical rather than formal.** Concretely, SINC-GCN is SIR-GCN plus an additive `Σ_w W_N h_w` term inside the nonlinearity (Eq. 15); benchmark gains over SIR-GCN are often within 1–2 absolute points (WikiCS 78.06 → 78.17, ogbn-arxiv 72.52 → 72.66, ogbg-molhiv 77.63 → 78.50). These are statistically significant under the authors' tests, but effect sizes (percentage improvement or Cohen's d), not just p-values, should be reported, and phrasings such as "novel theoretical paradigm" (Section 3.1) and "lays the foundation for the NCMP framework" should be toned down. Relatedly, Section 3.1.1 says SINC-GCN "retains the 1-WL upper bound" while capturing structural semantics "standard message-passing fundamentally ignores," but no formal separation from SIR-GCN is given; UniqueSignature is suggestive, not a WL-style impossibility theorem, and the biconnectivity-based expressivity line [5] should be cited in Section 2.4 alongside the already-cited expressivity references.
- **Synthetic evidence rests on a single, partly built-in probe.** UniqueSignature probes one failure mode (sum-to-target over neighbor weights), and since the closed-form optimal SINC-GCN parameters are essentially handed to the reader, the 1.00 accuracy is partly built-in. A second synthetic task (e.g., neighborhood-variance detection, neighborhood-majority-label, or neighborhood-cardinality-dependent classification) would support the claim that NCMP captures a *general* family of neighborhood-aggregate properties rather than just this one.
- **Parameter-budget confound in Claim 5 and missing context-aware baselines.** Table 4's `none` ablation is the right control, but `W_N` introduces extra parameters and the 100K budget forces SINC-GCN to use a *smaller* hidden dimension than baselines (Appendix A.2); whether an SIR-GCN with an equivalent-capacity extra linear path — matched in total parameters but without the set-level aggregation — closes the gap is not shown. A parameter-matched ablation would tighten the "it's neighborhood-context, not raw capacity" argument. Separately, Subgraph-GNNs and graph-transformer-style methods are scoped out of Table 3, but at least one representative at matched budget — [3] (ESAN) or [4] (GNN-AK) from the subgraph-GNN line, or [2] (GraphGPS) as the canonical local-MP + global-attention reference — belongs in Table 3 so the reader can calibrate what a cheap neighborhood-context term buys relative to a more expensive global-context model. [2] is the most direct foil for NCMP and is not currently cited.
- **Missing canonical reference for the aggregator and for the pattern-based / foundation-model lines.** The `Σ_w W_N h_w` construction in Eq. 14 is literally a Deep Sets aggregator on the neighborhood multiset; [1] is the foundation of the permutation-equivariance argument in Section 3.2 and should be cited around Eq. 14. Section 2.4 also misses two closely related threads: the pattern-based line [6, 7] is an *aggressive* response to the same diagnosis NCMP responds to — it breaks from MP entirely in favor of subgraph / pattern tokenization — and NCMP is the *conservative* alternative; positioning these side-by-side clarifies the efficiency-vs.-clean-break design trade-off. The computation-tree / task-tree line [8, 9] makes precise what "local structural semantics" means across graphs (the neighborhood set `{h_w : w ∈ N(u)}` is exactly a depth-1 task-tree) and gives the "generalize to unseen graphs" claim on p. 7 a concrete reference; [10] as a broader-landscape pointer and [11] as an efficiency-Pareto neighbor are optional additions.

---

## References

[1] *Deep Sets.* NeurIPS 2017.

[2] *Recipe for a General, Powerful, Scalable Graph Transformer (GraphGPS).* NeurIPS 2022.

[3] *Equivariant Subgraph Aggregation Networks (ESAN).* ICLR 2022.

[4] *From Stars to Subgraphs: Uplifting Any GNN with Local Structure Awareness (GNN-AK).* ICLR 2022.

[5] *Rethinking the Expressive Power of GNNs via Graph Biconnectivity.* ICLR 2023.

[6] *Beyond Message Passing: Neural Graph Pattern Machine (GPM).* ICML 2025.

[7] *Generative Graph Pattern Machine (G2PM).* NeurIPS 2025.

[8] *GFT: Graph Foundation Model with Transferable Tree Vocabulary.* NeurIPS 2024.

[9] *Towards Graph Foundation Models: Learning Generalities Across Graphs via Task-Trees (GIT).* ICML 2025.

[10] *Graph Foundation Models: A Comprehensive Survey.* arXiv:2505.15116, 2025.

[11] *Training MLPs on Graphs Without Supervision (SimMLP).* WSDM 2025.

**Audience:**

Yes

**Audience Explanation:**

See contributions

**Broader Impact Concerns:**

None. The work is a general architectural / theoretical contribution to GNN design, evaluated on standard public datasets. No dual-use or societal concerns are raised, and none need to be added.

**Claims And Evidence:**

Yes

**Claims Explanation:**

See contributions

**Requested Changes:**

* **Strengthen the expressivity story.** Either (a) add a formal separation result showing a graph family SINC-GCN distinguishes but SIR-GCN provably cannot, or (b) soften Section 3 / the abstract to match what is actually shown — that SINC-GCN generalizes SIR-GCN and empirically helps on tasks requiring neighborhood-aggregate reasoning. Either path, cite [5] in Section 2.4 alongside the already-cited expressivity references.
* **Add a second synthetic probe.** For example, neighborhood-majority-label classification, or distinguishing two graphs with identical 1-WL color histograms but different neighbor-feature moment distributions. This would substantially strengthen the case that NCMP is a *general* way to capture neighborhood context.
* **Report effect sizes alongside p-values in Table 3.** Percentage improvement or Cohen's d per dataset.
* **Add one context-aware non-one-hop baseline in Table 3.** GraphGPS [2] (small variant) or a subgraph-GNN ([3] or [4]) at matched parameter budget.
* **Add a parameter-matched ablation** replacing `W_N h_w` with an equivalent-capacity extra linear path in SIR-GCN, so Claim 5 is not confounded with raw parameter budget.
* **Close the related-work gaps.** Cite [1] around Eq. 14 for the Deep Sets foundation of the permutation-equivariance argument; cite [2] in Section 2.4 as the canonical local-MP + global-attention reference; cite [3] and [4] in Section 2.4 as the subgraph-GNN baselines; cite [6] and [7] in Section 2.4 alongside the already-cited subgraph / substructure references as the aggressive pattern-based alternatives that NCMP positions against; cite [8] and [9] in Section 3 (right after NCMP is introduced) or Section 4.2 (OGB transfer) to support the "depth-1 computation-tree" framing and the "generalize to unseen graphs" claim on p. 7; optionally cite [10] as a survey pointer and [11] in Appendix B as an efficiency-Pareto neighbor.

---

> ### Author Response · Authors · 2026-05-03
>
> We thank the reviewer for the insightful and thorough feedback. We have revised the manuscript to address the requested changes as follows:
>
> 1. **Strengthened Theoretical Discussion.** Section 3.1.1 has been substantially expanded to formally characterize the class of graph properties uniquely distinguishable by NCMP and the SINC-GCN instance. This involves continuous, relational feature dynamics and structural gating within the local one-hop neighborhood, supporting the empirical results in Section 4.
>
> 2. **Added Synthetic Datasets.** To provide an unbiased assessment of expressivity beyond the customized UniqueSignature dataset, we have integrated the standard EXP and CEXP synthetic probes into our evaluation. The results demonstrate that on 1-WL distinguishable instances, SINC-GCN significantly outperforms standard message-passing architectures by a substantial margin, frequently achieving perfect accuracy across multiple random seeds. This further underscores the practical utility of neighborhood-contextualization as a simple, highly scalable, and powerful inductive bias for localized message-passing.
>
> 3. **Reported Effect Sizes for Benchmark Results.** We have updated the benchmark datasets results by including Cohen's $d$ effect sizes. These metrics reveal that SINC-GCN largely achieves substantial performance gains beyond statistical significance, yielding large to huge effect sizes.
>
> 4. **Expanded Model Comparison for Benchmark Results.** To provide a more comprehensive evaluation, the benchmark datasets results have been expanded to incorporate advanced models (CWN, GraphGPS) evaluated under identical experimental set-up. Notably, the additional results demonstrate SINC-GCN achieving highly competitive or superior performance against the globally-attentive GraphGPS on datasets such as MNIST and CIFAR10.
>
> 5. **Added Ablation Results.** We acknowledge the importance of cleanly isolating the effect of neighborhood-contextualization, unconfounded by the raw parameter budget. However, injecting an equivalent-capacity "dummy" linear path into SIR-GCN to replace the neighborhood context creates mathematical redundancy, as this new linear transformation may be combined or "absorbed" into the existing $\boldsymbol{W_Q}$, rendering the model identical to the baseline. Consequently, we provided a dimension-matched ablation in Appendix B.3, matching the dimensionality of the hidden representation of SINC-GCN to the baseline SIR-GCN. The results remain largely robust, with the contextualized models outperforming the baseline. Nevertheless, as SINC-GCN explicitly introduces an additional weight $\boldsymbol{W_N}$ to extract the neighborhood context, increasing its overall parameters, and consequently resulting in worse performance in some datasets, which may be reasonably attributed to overfitting as the neighborhood context already provides a highly expressive inductive bias. Overall, the additional analyses further validate the expressivity of SINC-GCN with its neighborhood-contextualization, while also demonstrating its parameter efficiency.
>
> 6. **Strengthened Related Works.** We substantially expanded Section 2.4, incorporating the suggested works, to provide a more comprehensive discussion of related works. This section now includes dedicated subsections on: 1) Multiple Aggregators & Higher-Order Neighborhoods, 2) Subgraph-Enhanced GNNs & Biconnectivity, 3) Graph Transformers & Global Context, and 4) Pattern-Based & Graph Foundation Models. Furthermore, we added a dedicated subsection to formally articulate the architectural distinction between the NCMP framework and these advanced paradigms. Finally, a discussion of set-based aggregation methods was integrated into Section 3.1 to properly motivate and situate the formulation of SINC-GCN in relation to these established theories.
>
> 7. **Softened Novelty Claims.** We have comprehensively revised the manuscript to temper and soften our claims regarding novelty and contributions. Specifically, we now explicitly state that neighborhood-contextualization adapts and refines well-established concepts from set-based aggregation literature for the specific context of GNNs. With this as basis, the proposed NCMP framework functions as a practical generalization of the standard message-passing variant. Finally, SINC-GCN is presented as a simple, mathematically grounded method for its efficient parametrization and operationalization.

---

> ### Author Response · Authors · 2026-05-03
> **Additional Results**
>
> > **Table: Test Accuracy on EXP & CEXP.**
>
> | **Model** | **$\text{EXP}$** | **$\text{Corrupt}$** | **$\overline{\text{EXP}}$** | **$\text{CEXP}$** |
> | :-------------- | :----------------------: | :--------------------------: | :---------------------------------: | :-----------------------: |
> | GCN             |      50.00 ± 0.00      |        64.50 ± 3.17        |            50.00 ± 0.00            |       57.25 ± 1.58       |
> | GraphSAGE       |      50.00 ± 0.00      |        50.00 ± 0.00        |            50.00 ± 0.00            |       50.00 ± 0.00       |
> | GATv2           |      50.00 ± 0.00      |        61.17 ± 12.47        |            50.00 ± 0.00            |       55.58 ± 6.24       |
> | GIN             |      50.00 ± 0.00      |        53.33 ± 7.92        |            50.00 ± 0.00            |       51.67 ± 3.96       |
> | SIR-GCN         |      50.00 ± 0.00      |        51.83 ± 5.50        |            50.00 ± 0.00            |       50.92 ± 2.75       |
> | SINC-GCN        |      50.00 ± 0.00      |        85.50 ± 20.62        |            50.00 ± 0.00            |      67.75 ± 10.31      |
> | PNA             |      50.00 ± 0.00      |        99.83 ± 0.50        |            50.00 ± 0.00            |       74.92 ± 0.25       |
> | EGC-S           |      50.00 ± 0.00      |        50.00 ± 0.00        |            50.00 ± 0.00            |       50.00 ± 0.00       |
> | EGC-M           |      50.00 ± 0.00      |        65.83 ± 22.06        |            50.00 ± 0.00            |      57.92 ± 11.03      |
>
> > **Table: Test Performance on Benchmark Datasets.**
>
> | **Model** | **WikiCS (↑)** | **PATTERN (↑)** | **CLUSTER (↑)** | **MNIST (↑)** | **CIFAR10 (↑)** | **ZINC (↓)** | **ogbn-arxiv (↑)** | **ogbg-molhiv (↑)** |
> | :-------------- | :-------------------: | :--------------------: | :--------------------: | :------------------: | :--------------------: | :-----------------: | :-----------------------: | :------------------------: |
> | SIR-GCN         |     78.06 ± 0.66     |     85.75 ± 0.03     |     63.35 ± 0.19     |    97.90 ± 0.08    |     71.98 ± 0.40     |   0.278 ± 0.024   |       72.52 ± 0.16       |       77.63 ± 0.84       |
> | SINC-GCN        |     78.17 ± 0.68     |     85.79 ± 0.02     |     63.51 ± 0.15     |    98.28 ± 0.05    |     73.37 ± 0.41     |   0.256 ± 0.006   |       72.66 ± 0.09       |       78.50 ± 1.23       |
> | Cohen's d       |        0.1691        |         1.5104         |         0.8817         |        5.1610        |         3.4732         |       1.0801       |          0.9900          |           0.8894           |
> | CWN             |           -           |           -           |           -           |          -          |           -           |   0.139 ± 0.008   |             -             |       80.55 ± 1.04       |
> | GraphGPS        |           -           |           -           |           -           |    98.05 ± 0.13    |     72.30 ± 0.36     |          -          |             -             |             -             |
>
> > **Table: Dimension-Matched Ablation & Sensitivity Analyses.**
>
> | **Aggregator ⨂** | **WikiCS (↑)** | **PATTERN (↑)** | **CLUSTER (↑)** | **MNIST (↑)** | **CIFAR10 (↑)** | **ZINC (↓)** | **ogbn-arxiv (↑)** | **ogbg-molhiv (↑)** |
> | :---------------------- | :-------------------: | :--------------------: | :--------------------: | :------------------: | :--------------------: | :-----------------: | :-----------------------: | :------------------------: |
> | none                    |     78.06 ± 0.66     |     85.75 ± 0.03     |     63.35 ± 0.19     |    97.90 ± 0.08    |     71.98 ± 0.40     |   0.278 ± 0.024   |       72.52 ± 0.16       |       77.63 ± 0.84       |
> | sum                     |       FP Error       |     85.72 ± 0.02     |     59.80 ± 0.13     |    97.85 ± 0.09    |     70.46 ± 1.20     |   0.267 ± 0.010   |       70.58 ± 0.74       |       76.40 ± 1.36       |
> | max                     |     76.30 ± 1.80     |     85.63 ± 0.08     |     62.58 ± 0.37     |    98.08 ± 0.10    |     73.06 ± 0.32     |   0.249 ± 0.006   |       71.61 ± 0.18       |       75.69 ± 0.80       |
> | mean                    |     78.15 ± 0.63     |     85.78 ± 0.03     |     63.48 ± 0.17     |    98.25 ± 0.13    |     73.67 ± 0.47     |   0.278 ± 0.026   |       72.25 ± 0.12       |       75.74 ± 1.48       |
> | sym. mean               |     78.15 ± 0.65     |     85.78 ± 0.02     |     63.67 ± 0.19     |    98.23 ± 0.07    |     73.55 ± 0.64     |   0.259 ± 0.013   |       72.49 ± 0.08       |       77.54 ± 2.10       |

---

> ### Author Response · Authors · 2026-05-03
> **Theoretical Discussion on Expressivity**
>
> > With standard (non-attentional) message-passing, messages $\boldsymbol{\psi_{u,v}}$ are strictly independent of $\boldsymbol{h_w}$, for $w \in \mathcal{N}(u) \setminus \{v\}$. Hence, any modulation to $\boldsymbol{\psi_{u,v}}$ based on the neighborhood feature distribution may only occur in the subsequent layer, after the set of messages has been irreversibly compressed into a single vector representation.
>
> > For attention head $h$ in multi-head GAT, $\boldsymbol{\psi_{u,v}^{(h)}} = \alpha_{u,v}^{(h)} \cdot \boldsymbol{W^{(h)}} \boldsymbol{h_v}$. Hence, for $w \in \mathcal{N}(u) \setminus \{v\}$, $\frac{\partial}{\partial \boldsymbol{h_w}} \boldsymbol{\psi_{u,v}^{(h)}} = \boldsymbol{W^{(h)}} \boldsymbol{h_v} \left(\nabla_{\boldsymbol{h_w}} \alpha_{u,v}^{(h)}\right)^\top$, which is strictly rank-1. Hence, any change in $\boldsymbol{h_w}$ can only **scale** $\boldsymbol{\psi_{u,v}^{(h)}}$ along its **original feature space direction**, and not project it into new semantic directions.
>
> > By directly injecting $\boldsymbol{N_u}$ into $\boldsymbol{\psi_{u,v}}$, SINC-GCN permits conditional message transformation based on the neighborhood feature distribution, with a non-zero, full-rank Jacobian $\frac{\partial}{\partial \boldsymbol{h_w}} \boldsymbol{\psi_{u,v}}$, for $w \in \mathcal{N}(u) \setminus \{v\}$. Consequently, any change in $\boldsymbol{h_w}$ can fundamentally alter $\boldsymbol{\psi_{u,v}}$ along **any feature space direction**. Hence, the class of graph properties uniquely distinguishable by NCMP and the SINC-GCN instance involves continuous, relational feature dynamics and structural gating within the local one-hop neighborhood.

---

### Review · Reviewer_4m9K · 2026-05-08

**Summary Of Contributions:**

This paper introduces a new implementation variant for GNN (Neighborhood-Contextualized Message-Passing (NCMP)) that aims to capture meaningful structural relationships while preserving computational efficiency.

As a message-passing approach, its main contribution is enhancing representational power of message-passing GNN, by ensuring messages are fully contextualized within the broader local nodes neighborhood. Its implementation, SINC-GCN aims at striking an optimal balance between model expressivity and computational efficiency.

**Additional Comments:**

The paper is remarkably balanced between fundamental aspects and results presentation. It's only missing a few additional paragraphs centered on applications, both in terms of benchmark results' interpretation and in terms of further work.

**Audience:**

Yes

**Audience Explanation:**

Although the paper covers a fairly specialized topic, it does so in a way that would be of interest to a broader readership than simply colleagues working on GNN. This potential outreach could be further strengthened by expanding the discussion on the kind of applications that would benefit most from Neighborhood-Contextualized Message Passing.

**Claims And Evidence:**

Yes

**Claims Explanation:**

The paper contains enough mathematical details to substantiate the approach without obscuring the overall presentation. It introduces GIN and its grounding in Weisfeiler-Lehman, then attentional and message-passing variants which lead into a justification of SINC-GCN on an efficiency basis. These sections together with the introductory classification of GNN approaches is one of the strong points of the paper.

Table 3 provides comprehensive information on SINC-GCN performance on a wide range of benchmarks with different problems characteristics (e.g. WikiCS, CIFAR10 and ZINC) with results varying from non-significant to several percentage points (significant). Although these results tend to be well supported by the subsequent ablation study (e.g. for ZINC and CIFAR10) the effect size remains moderate.

**Requested Changes:**

Throughout the paper there are rather implicit reference to application areas (‘quantum mechanical simulations’ in 2.3, and of course, chemistry via the ZINC dataset). Since a broad range of applications is mentioned in the introduction and datasets vary accordingly it would be really useful to include a discussion of those applications that would benefit more from node contextualization.

Considering that the magnitude of improvements from baseline varies from moderate to significant it seems reasonable to complement the results section some qualitative analysis of which application areas could most benefit from the specific features of this novel approach (namely, contextualized message passing and capturing structural relationships).

For the same reasons, since the paper itself refers to the current implementation as a conceptual proof, it might be worth expanding the section on further work to give more precise directions (one item already refers to applications, but the potential alternative to softmax attention could be developed)

---

> ### Author Response · Authors · 2026-05-09
>
> We thank the reviewer for the thoughtful feedback. We have revised the manuscript to address the requested changes as follows:
>
> 1. **Detailed Discussion on Applications.** In Section 1 Paragraphs 1, 2, and 4, we have provided a more detailed discussion of how neighborhood context becomes critical for complex GNN applications like financial systems, social networking sites, and molecules.
>
> 2. **Added Qualitative Analysis of Benchmark Results.** We have provided a qualitative analysis of the benchmark results in Section 4.2 and Appendix B.2. We first described how standard one-hop localized GNNs already capture sufficient local structural semantics for node property prediction tasks. Meanwhile, graph property prediction tasks require aggregating node-level information across the entire graph, making localized neighborhood contexts critical for preserving crucial information for downstream tasks.
>
> 3. **Expanded Future Works.** We have significantly expanded the Conclusion section to provide more concrete directions for future works to build upon the current paper. These include applying SINC-GCN on critical domains, exploring more sophisticated NCMP parametrizations integrating advanced GNN paradigms, and formally developing neighborhood-contextualization as a robust and direct alternative to softmax attention in Transformer by extending the Jacobian analysis provided in Section 3.1.1.

---

### Decision · Action_Editor_MpKG · 2026-06-08

**Recommendation:** Accept with minor revision

**Audience:**

Yes

**Audience Explanation:**

Yes, people in GNN community will find the paper interesting

**Claims And Evidence:**

Yes

**Claims Explanation:**

Claim are well-supported, especially after the revision and additional experiments have been added.

It would be nice to add more details about possible applications. In particular, at least for one of the application it would be nice to see an extended and more comprehensive description.

---

> ### Author Response · Authors · 2026-06-20
> **Camera Ready Submission**
>
> We would like to express our gratitude to the reviewers and action editor for their time and insightful feedback. We have submitted the Camera Ready Revision and provided a more comprehensive discussion in the Conclusion section on applying the NCMP framework to LLMs and the Transformer architecture.